# Training-Free Bayesian Filtering with Generative Emulators

**Thomas Savary** [1]   **François Rozet** [1]   **Gilles Louppe** [1]

## Abstract

Bayesian filtering is a well-known problem that aims to estimate plausible states of a dynamical system from observations. Among existing approaches to solve this problem, particle filters are theoretically exact for non-linear dynamics and observations, but suffer from poor scalability in high dimensions. In this work, we show that diffusion-based emulators of dynamical systems can be used to implement, without additional training, an optimal variant of particle filters that has remained largely unexplored due to implementation challenges with classical numerical solvers. Experiments on nonlinear chaotic systems, including atmospheric dynamics, demonstrate that the proposed approach successfully scales particle filtering to high-dimensional settings.

## 1. Introduction

Numerical simulation of dynamical systems is a central tool in science, enabling the exploration and prediction of complex phenomena that are not accessible to direct experimentation. Traditionally, it relies on modeling the dynamics using partial differential equations, which are then solved with numerical methods (Lorenz, 1963; Chorin, 1968; Hairer et al., 2000). This approach requires significant modeling and numerical effort, and becomes computationally expensive for large dynamical systems.

Recently, deep neural networks have emerged as a compelling alternative, achieving competitive accuracy at substantially lower computational cost (Tompson et al., 2017; Lam et al., 2023). In particular, generative models (Song et al., 2021b; Lipman et al., 2023; Albergo et al., 2025) attract growing interest due to their ability to capture high-dimensional, multimodal distributions, making them promising candidates for efficient simulation of dynamical systems (Chen et al., 2024; Rozet et al., 2025).

[1]SAIL, Montefiore institute, University of Liège, Belgium. Correspondence to: Thomas Savary <tsavary@uliege.be>.

*Proceedings of the 43rd International Conference on Machine Learning*, Seoul, South Korea. PMLR 306, 2026. Copyright 2026 by the author(s).

A frequently overlooked aspect of these methods is their dependence on an initial condition to start the simulation. The latter is particularly important for chaotic systems, where small deviations in the initial state grow exponentially over time (Devaney, 2003). Since, in most real-world settings, it is difficult to accurately estimate the state of the system at a given time, a common practice, known as data assimilation, is to use a set of observations to infer the most probable states (Carrassi et al., 2018). Various algorithms have been proposed to tackle this problem (Lorenc, 1986; Evensen, 2009), and this paper contributes to ongoing efforts to adapt and design data assimilation algorithms with generative models (Rozet & Louppe, 2023; Huang et al., 2024).

### 1.1. Problem statement

We consider a discrete-time Markovian dynamical system with unknown state $x^k$ at time step $k$. The goal, known as Bayesian filtering (Särkkä, 2013), is to estimate $x^k$ from past and current observations $y^{1:k} = (y^1, y^2, \ldots, y^k)$, that is, to approximate the posterior distribution $p(x^k \mid y^{1:k})$. For example, in weather forecasting, $x^k$ represents the state of the atmosphere at a given time, while observations $y^{1:k}$ come from ground weather stations and satellites (Brousseau et al., 2025). We further assume access to a generative emulator of the dynamics, in the form of a diffusion model (see Section 2.3), which allows generating probable future states $x^{k+1}$ from a given current state $x^k$, that is, drawing samples from $p(x^{k+1} \mid x^k)$.

### 1.2. Contributions

In this work, we propose a simple but effective method that adapts diffusion emulators to perform Bayesian filtering without additional training. The method is mathematically grounded and establishes an elegant connection between generative models and particle filters (van Leeuwen et al., 2019), in particular the fully adapted particle filter (Petetin & Desbouvries, 2013). We show that, contrary to common belief (Snyder et al., 2008), particle filters can be applied in high-dimensional problems when combined with generative models, even with relatively few particles. To illustrate this, we apply our method on GenCast, a global diffusion-based emulator of the atmosphere (Price et al., 2025). The code is available at https://github.com/ThomasSavary08/FA-APF.

## 2. Preliminaries

### 2.1. Data assimilation and Bayesian filtering

As explained earlier, the goal of data assimilation algorithms is to estimate the state $x^k$ of a dynamical system at time $k$ from a set $\mathcal{I}$ of observations $\{y^i\}_{i \in \mathcal{I}}$, a model of the system dynamics, and a model for the observations. Formally, following the notations of Carrassi et al. (2018), we define the dynamics as

$$x^{k+1} = \mathcal{M}(x^k, \lambda) + \eta^{k+1} \sim p(x^{k+1} \mid x^k), \quad (1)$$

where $\mathcal{M}$ is a transition model, $\lambda$ the parameters of the model, and $\eta^{k+1}$ a stochastic additive term intended to represent the error between the model prediction and the true unknown process initialized from the perfect initial condition. This term accounts for the cumulative effect of errors in the parameters $\lambda$, errors in the transition model $\mathcal{M}$, and the effect of unresolved scales (Carrassi et al., 2018). In Section 3, we assume access to a diffusion model that directly samples from $p(x^{k+1} \mid x^k)$.

For the observations, we assume that the measurement process can be modeled by an operator $\mathcal{H} : \mathbb{R}^n \longrightarrow \mathbb{R}^d$ mapping the state space to the observation space, leading to the following formulation

$$y^k = \mathcal{H}(x^k) + \varepsilon^k. \quad (2)$$

Similarly to the model error, the additive stochastic term $\varepsilon^k$ accounts for the instrumental error of observing devices (such as satellites), and deficiencies in the formulation of the observation operator itself (Carrassi et al., 2018). As commonly assumed in other data assimilation works (Rozet & Louppe, 2023; Bao et al., 2024; Andry et al., 2025), we model this additive term as a zero-mean Gaussian with covariance $\Sigma_y$, leading to a Gaussian distribution for the observations

$$y^k \sim \mathcal{N}(\mathcal{H}(x^k), \Sigma_y). \quad (3)$$

Depending on the set of observations used to estimate the system state, we commonly distinguish two main problems in data assimilation. When past, current, and future observations are considered, the problem is referred to as Bayesian smoothing (Särkkä, 2013). Solutions to this problem are primarily used to construct reanalysis datasets such as ERA5 (Hersbach et al., 2020). When only past and current observations are used, the problem reduces to Bayesian filtering, as introduced in the problem statement. Solutions to this problem are mainly used operationally to determine initial conditions for simulations (Brousseau et al., 2025).

### 2.2. Particle filters

Particle filters are a family of data assimilation algorithms designed to estimate the filtering distribution $p(x^k \mid y^{1:k})$

by a discrete probability measure $\mu^k = \sum_{i=1}^{N} w_i^k \delta_{x_i^k}$ that converges weakly

$$\sum_{i=1}^{N} w_i^k g(x_i^k) \underset{N \to +\infty}{\longrightarrow} \int g(x^k) p(x^k \mid y^{1:k}) \mathrm{d}x^k, \quad (4)$$

where $x_i^k$ are the particles at time $k$, $w_i^k$ the associated weights, $y^{1:k}$ the observations and $g$ any continuous and bounded function (van Leeuwen et al., 2019).

Particle filters are derived from the Bayesian filtering recursion: at each time step $k$, the next filtering distribution is obtained through a prediction and update step

$$p(x^{k+1} \mid y^{1:k}) = \int p(x^{k+1} \mid x^k) p(x^k \mid y^{1:k}) \mathrm{d}x^k, \quad (5)$$

$$p(x^{k+1} \mid y^{1:k+1}) \propto p(y^{k+1} \mid x^{k+1}) p(x^{k+1} \mid y^{1:k}). \quad (6)$$

In practice, the prediction step is done by sampling the next particles from a proposal distribution $q(x^{k+1} \mid x^k, y^{k+1})$ that represents any transition distribution conditioned on the current state $x^k$ and, optionally, the next observation $y^{k+1}$. Weights are then updated to correct the mismatch between the predicted and true posterior distribution

$$\hat{w}_i^{k+1} = \frac{p(y^{k+1} \mid x_i^{k+1}) p(x_i^{k+1} \mid x_i^k)}{q(x_i^{k+1} \mid x_i^k, y^{k+1})} \times w_i^k, \quad (7)$$

and subsequently normalized to obtain a valid probability measure

$$w_i^{k+1} = \frac{\hat{w}_i^k}{\sum_{j=1}^{N} \hat{w}_j^k}. \quad (8)$$

Unlike classical approaches such as ensemble Kalman filters (Evensen, 2009) or variational methods (Le Dimet & Talagrand, 1986), particle filters offer the theoretical advantage of handling non-linear transition and observation models, which arise in most dynamical systems of interest. However, in high-dimensional systems, particle filters suffer from the curse of dimensionality (Van Leeuwen, 2015). This phenomenon, known as degeneracy, corresponds to the situation where only a small subset of particles have non-negligible weights. It is due to the dimension of the observation space: the higher this dimension, the more peaked the likelihood is, and the more unlikely it is for the majority of particles to end up close to the observation (Van Leeuwen, 2015).

To alleviate this issue, particle filtering algorithms commonly introduce a resampling step, which preserves particle diversity and resets the weights. Also, it has been shown in the literature that using the optimal proposal $q(x^{k+1} \mid x^k, y^{k+1}) = p(x^{k+1} \mid x^k, y^{k+1})$ during the sampling step minimizes the variance of the weights and, consequently, reduces degeneracy (Snyder et al., 2015). However, the use of the optimal proposal is rarely feasible in practice, as it is difficult to implement for standard simulators.

## 2.3. Diffusion models for probabilistic forecasting

Diffusion models (Ho et al., 2020; Song et al., 2021b), are a class of generative models to sample plausible data from a distribution $p(x)$ of interest. Formally, adapting the formulation of Song et al. (2021b), samples $x \in \mathbb{R}^n$ from $p(x)$ are progressively perturbed through a diffusion process expressed as a stochastic differential equation (SDE)

$$dx_t = f_t x_t dt + g_t dw_t, \tag{9}$$

where $f_t \in \mathbb{R}$ is the drift coefficient, $g_t \in \mathbb{R}_+$ the diffusion coefficient, $w_t \in \mathbb{R}^n$ a standard Wiener process, and $x_t \in \mathbb{R}^n$ the perturbed sample at time $t \in [0, 1]$. Because the SDE is linear, the perturbation kernel from $x$ to $x_t$ is Gaussian and takes the form

$$p_t(x_t \mid x) = \mathcal{N}(x_t \mid \alpha_t x, \Sigma_t), \tag{10}$$

where $\alpha_t$ and $\Sigma_t = \sigma_t^2 I$ are derived from $f_t$ and $g_t$ (Anderson, 1982). Crucially, the forward SDE (9) has an associated family of reverse SDEs (Anderson, 1982)

$$dx_t = \left[ f_t x_t - \frac{1 + \eta^2}{2} g_t^2 \nabla_{x_t} \log p_t(x_t) \right] dt + \eta g_t dw_t, \tag{11}$$

where $\eta$ is a parameter controlling stochasticity. In simpler words, we can draw noise samples from $p(x_1) \approx \mathcal{N}(0, \Sigma_1)$ and gradually remove the noise to obtain samples from $p(x)$ by simulating Equation (11) from $t = 1$ to $t = 0$ using an appropriate discretization scheme (Zhang & Chen, 2023; Song et al., 2021a).

In practice, the score function $\nabla_{x_t} \log p_t(x_t)$ in Equation (11) is unknown, but can be approximated by a neural network $d_\theta$ called denoiser and trained with the following loss function

$$\mathcal{L}(\theta) = \mathbb{E}_{x,t,x_t} \left[ \lambda_t \| x - d_\theta(t, x_t) \|_2^2 \right]. \tag{12}$$

In fact, the optimal denoiser is the mean $\mathbb{E}[x \mid x_t]$ of $p(x \mid x_t)$, and is linked to score function through the first order Tweedie's formula (see Appendix A)

$$\nabla_{x_t} \log p_t(x_t) = \Sigma_t^{-1}(\alpha_t \mathbb{E}[x \mid x_t] - x_t). \tag{13}$$

Given a dataset $\{(x_i^k, x_i^{k+1})\}_{i \in \mathcal{I}}$ of successive states of a dynamical system, we can adapt this paradigm to construct a generative emulator of the dynamic. To do so, we reformulate the training objective given in Equation (12) as

$$\mathbb{E}_{(x^k, x^{k+1}), t, x_t^{k+1}} \left[ \lambda_t \| x^{k+1} - d_\theta(t, x^k, x_t^{k+1}) \|_2^2 \right]. \tag{14}$$

In this case, the optimal denoiser is the conditional mean $\mathbb{E}[x^{k+1} \mid x_t^{k+1}, x^k]$ and is linked to the conditional score function $\nabla_{x_t^{k+1}} \log p_t(x_t^{k+1} \mid x^k)$ through Equation (13). Using this score in Equation (11) yields samples from $p(x^{k+1} \mid x^k)$, thereby enabling autoregressive emulation of the system dynamics. In the remainder of this work, we assume access to a diffusion-based emulator for a given dynamical system.

## 3. Method

As outlined in the introduction, the objective is to estimate the Bayesian filtering distribution $p(x^k \mid y^{1:k})$ using a diffusion-based model trained for sampling the transition distribution $p(x^{k+1} \mid x^k)$ of a given dynamical system. We show that such generative emulators can be directly used, without additional training, to implement the fully adapted auxiliary particle filter (Petetin & Desbouvries, 2013).

### 3.1. The Fully-Adapted Auxiliary Particle Filter

The fully adapted auxiliary particle filter (FA-APF) is a particle filtering algorithm designed to approximate the Bayesian filtering distribution (Petetin & Desbouvries, 2013). It is summarized in Algorithm 1, where $p(x^0)$ denotes the prior distribution of the initial state, $N$ the number of particles, and $K$ the total number of time steps. The parameters $\alpha$, $N_{\text{thr}}^{\min}$, $N_{\text{thr}}^{\max}$ are introduced below.

---

**Algorithm 1** Fully-Adapted Auxiliary Particle Filter

---

1: **Inputs:** $p(x^0)$, $N$, $N_{\text{thr}}^{\min,\max}$, $\alpha$, $K$
2: $x_i^0 \sim p(x^0)$
3: $w_i^0 \leftarrow 1/N$
4: **for** $k = 0$ **to** $K - 1$ **do**
5: $\quad \mu_i^{k+1} \leftarrow \mathbb{E}[x^{k+1} \mid x_i^k]$
6: $\quad$ **while** $N_{\text{eff}}$ not in $[N_{\text{thr}}^{\min}, N_{\text{thr}}^{\max}]$ **do**
7: $\quad\quad$ update/initialize $\alpha$
8: $\quad\quad \hat{w}_i^{k+1} \leftarrow \left[ p(y^{k+1} \mid \mu_i^{k+1}) \right]^\alpha$
9: $\quad\quad w_i^{k+1} \leftarrow \hat{w}_i^{k+1} / \sum_{j=1}^N \hat{w}_j^{k+1}$
10: $\quad\quad N_{\text{eff}} \leftarrow 1 / \sum_{i=1}^N (w_i^{k+1})^2$
11: $\quad$ **end while**
12: $\quad a_i^{k+1} \sim \text{Cat}(\{w_i^{k+1}\}_{1 \leq i \leq N})$
13: $\quad x_i^{k+1} \sim p(x^{k+1} \mid x_{a_i^{k+1}}^k, y^{k+1})$
14: **end for**
15: **Return** $\mu_x^k = \frac{1}{N} \sum_{i=1}^N \delta_{x_i^k}$ for all $k \in [1, K]$

---

Unlike most particle filters used in practice, FA-APF first computes the particle weights (lines 5–10 of Algorithm 1), and then propagates the particles to the next time step using these weights together with the optimal proposal $q(x^{k+1} \mid x^k, y^{k+1}) = p(x^{k+1} \mid x^k, y^{k+1})$ (lines 12-13). This choice minimizes the variance of the weights (Snyder et al., 2015) and therefore reduces particle degeneracy.

To further control degeneracy, following a strategy commonly used in data assimilation methods (Hunt et al., 2007), we introduce an inflation coefficient $\alpha$ and a control interval $[N_{\text{thr}}^{\min}, N_{\text{thr}}^{\max}]$ on the effective sample size, defined as the number of particles with non-negligible weights among the $N$ particles. While this introduces a bias in the approximation of the filtering distribution, it ensures that the variance of particles is non-zero at each time step $k$.

## 3.2. Sampling from the optimal proposal

To apply the FA-APF, we must sample from the optimal proposal distribution $p(x^{k+1} \mid x^k, y^{k+1})$, which is generally infeasible for standard simulators. However, as shown by Chen et al. (2026), Savary et al. (2025) and Andrae et al. (2025), this distribution can be accessed using diffusion emulators.

The key idea is to incorporate the observation $y^{k+1}$ into the score when solving the reverse diffusion Equation (11), that is, using the score $\nabla_{x_t^{k+1}} \log p_t(x_t^{k+1} \mid x^k, y^{k+1})$ of the posterior. Thanks to Bayes' rule, this score can be decomposed as

$$s_t^{x,y}(x_t^{k+1}, x^k, y^{k+1}) = s_t^x(x_t^{k+1}, x^k) + s_t^y(x_t^{k+1}, x^k, y^{k+1}), \tag{15}$$

where $s_t^{x,y}$ denotes the score of the posterior, $s_t^x$ the score of the prior, and $s_t^y$ the score of the likelihood. Since the score of the prior is already available from the trained denoiser of the emulator through Equation (13), the only unknown quantity that remains to be computed is the score $s_t^y(x_t^{k+1}, x^k, y^{k+1})$ of the likelihood.

To do so, we use moment matching posterior sampling (MMPS, Rozet et al., 2024), a method that approximates the likelihood by

$$p(y^{k+1} \mid x_t^{k+1}, x^k) \tag{16}$$

$$= \int p(y^{k+1} \mid x^{k+1}) p(x^{k+1} \mid x_t^{k+1}, x^k) \mathrm{d}x^{k+1} \tag{17}$$

$$\approx \int p(y^{k+1} \mid x^{k+1}) q(x^{k+1} \mid x_t^{k+1}, x^k) \mathrm{d}x^{k+1} \tag{18}$$

$$= \mathcal{N}(y^{k+1} \mid \mathcal{H}(\mathrm{m}), \Sigma_y + \mathrm{H V H}^\top), \tag{19}$$

where $q(x^{k+1} \mid x_t^{k+1}, x^k)$ is the density of Gaussian random variable with mean $\mathrm{m} = \mathbb{E}[x^{k+1} \mid x_t^{k+1}, x^k]$ and covariance $\mathrm{V} = \mathbb{V}[x^{k+1} \mid x_t^{k+1}, x^k]$, and H the Jacobian matrix of the observation operator. Assuming that the derivative of $\mathbb{V}[x^{k+1} \mid x_t^{k+1}, x^k]$ with respect to $x_t^{k+1}$ is negligible, we can then estimate the score of the likelihood as

$$\nabla_{x_t^{k+1}}^\top(\mathrm{m}) \mathrm{H}^\top (\Sigma_y + \mathrm{H V H}^\top)^{-1} \left[ y^{k+1} - \mathcal{H}(m) \right]. \tag{20}$$

The covariance matrix $\mathrm{V} = \mathbb{V}[x^{k+1} \mid x_t^{k+1}, x^k]$ is unknown a priori, but can be computed with the denoiser using the second-order Tweedie's formula (see Appendix A) as

$$\mathbb{V}[x^{k+1} \mid x_t^{k+1}, x^k] = \alpha_t^{-1} \Sigma_t \nabla_{x_t^{k+1}}^\top(\mathrm{m}). \tag{21}$$

Since the Jacobian $\nabla_{x_t^{k+1}}^\top d_\theta(t, x^k, x_t^{k+1}) \in \mathbb{R}^{n \times n}$ is intractable in high dimension and $(\Sigma_y + \mathrm{H V H}^\top)$ is symmetric positive definite, we consider only implicit access to the covariance matrix via automatic differentiation and we solve the linear system in Equation (20) using a linear solver (Saad & Schultz, 1986; van der Vorst, 1992).

Putting all these elements together, we can finally compute the score of the posterior and plug it into Equation (11) to generate samples from the optimal proposal $p(x^{k+1} \mid x^k, y^{k+1})$. The resulting procedure is summarized in Algorithm 2.

---

**Algorithm 2** Sampling from the optimal proposal

**Inputs:** $x^k$, $y^{k+1}$, $d_\theta$, $\Delta_t$, Solver

$x_{t=1}^{k+1} \sim \mathcal{N}(0, \Sigma_1)$

**for** $t$ in $[1, 1 - \Delta_t, \cdots, \Delta_t]$ **do**

$\quad s_x \leftarrow \Sigma_t^{-1} \left( \alpha_t d_\theta(t, x^k, x_t^{k+1}) - x_t^{k+1} \right)$ (Eq. 13)

$\quad s_y \leftarrow \mathrm{MMPS}(t, x_t^{k+1}, x^k, y^{k+1})$ (Eq. 20)

$\quad s_{x,y} \leftarrow s_x + s_y$ (Eq. 15)

$\quad x_{t-\Delta_t}^{k+1} \leftarrow \mathrm{Solver}(t, \Delta_t, x_t^{k+1}, s_{x,y})$ (Eq. 11)

**end for**

**Return** $x_{t=0}^{k+1}$

---

## 3.3. Computing weights

The final ingredient required to apply FA-APF is the computation of particle weights. When using the optimal proposal $q(x^{k+1} \mid x^k, y^{k+1}) = p(x^{k+1} \mid x^k, y^{k+1})$, the unnormalized weights in Equation (7) simplify to

$$\hat{w}_i^{k+1} = p(y^{k+1} \mid x_i^k) \times w_i^k, \tag{22}$$

where $w_i^k$ denotes the normalized weight of particle $i$ at time step $k$.

This quantity is not directly available, since the observation $y^{k+1}$ does not depend explicitly on $x^k$. We therefore approximate the transition distribution $p(x^{k+1} \mid x^k)$ by a Dirac mass at its conditional mean $\mathbb{E}[x^{k+1} \mid x^k]$ (Billingsley, 1995). This yields

$$p(y^{k+1} \mid x_i^k) \tag{23}$$

$$= \int p(y^{k+1} \mid x^{k+1}) p(x^{k+1} \mid x_i^k) \mathrm{d}x^{k+1} \tag{24}$$

$$\approx p(y^{k+1} \mid \mathbb{E}[x^{k+1} \mid x_i^k]) \tag{25}$$

$$= \mathcal{N}(\mathcal{H}(\mathbb{E}[x^{k+1} \mid x_i^k]), \Sigma_y). \tag{26}$$

For diffusion emulators, the conditional mean $\mathbb{E}[x^{k+1} \mid x_i^k]$ can be directly computed with one call to the trained denoiser as

$$\mathbb{E}[x^{k+1} \mid x^k] \underset{\varepsilon \sim \mathcal{N}(0, I)}{=} \mathbb{E}[x^{k+1} \mid x^k, \sigma_1 \varepsilon] \tag{27}$$

$$\approx d_\theta \left( x_{t=1}^{k+1} = \sigma_1 \varepsilon, x^k, t = 1 \right). \tag{28}$$

This approximation provides an efficient estimate of the particle weights and corresponds to lines 5–10 of Algorithm 1.

# 4. Experiments

## 4.1. Lorenz'63 System

We first evaluate our method on Lorenz'63 (L63), a simple yet widely used system to assess filtering performance. In one of its stochastic formulation (Chekroun et al., 2011), this system evolves according to

$$\begin{cases} \dot{x} = s(y - x) + x\sigma dw_t, \\ \dot{y} = x(r - z) - y + y\sigma dw_t, \\ \dot{z} = xy - bz + z\sigma dw_t, \end{cases} \quad (29)$$

where $(s, r, b)$ are the parameters of the dynamic, and $\sigma$ a parameter controlling the level of stochasticity. We set $(s = 10, r = 28, b = 8/3)$ to ensure a chaotic regime, typical of the systems encountered in practice, and $\sigma = 0.25$.

Numerical integration is performed using a Milstein scheme (Gelbrich & Römisch, 1995) with a time step of $10^{-3}$, and we collect snapshots of the system states every 0.5 time unit to create a dataset of successive states $\{(x_i^k, x_i^{k+1})\}_{i \in \mathcal{I}}$. The latter contains 10000 trajectories of 100 snapshots each, and is used to train a denoiser with the loss of Equation (14).

The denoiser is a residual network (He et al., 2016) with six hidden layers and follows the formalism of Karras et al. (2022). In particular, we adopt the same noise scheduling, with $\alpha_t = 1$ and $\sigma_t$ given by

$$\sigma_t = \left( \sigma_{\max}^{1/\rho} + (1 - t) \times \left( \sigma_{\min}^{1/\rho} - \sigma_{\max}^{1/\rho} \right) \right)^{\rho}, \quad (30)$$

where $\rho = 7$, $\sigma_{\min} = 10^{-3}$ and $\sigma_{\max} = 10^3$. The denoiser is trained for 10 epochs and, at inference time, we use a DDIM sampler (Song et al., 2021a) with 64 steps to solve the reverse diffusion Equation (11) and generate samples from the optimal proposal (see Algorithm 2).

We study the ability of our method to approximate the filtering distribution as a function of the number of particles ($N$ in Algorithm 1), also called number of ensemble members in the classical data assimilation literature (Carrassi et al., 2018). This analysis is essential for practical applications, as high-dimensional systems typically prevent the use of large ensembles because of computational cost (Leutbecher, 2019). We consider observations of the first and last component of the system, corrupted by Gaussian noise with standard deviation 0.25. This formally corresponds to define the observation $y^k$ in Equation (2) as

$$y^k = \underbrace{\begin{bmatrix} 1 & 0 & 0 \\ 0 & 0 & 1 \end{bmatrix} \begin{bmatrix} x \\ y \\ z \end{bmatrix}}_{\mathcal{H}(x^k)} + \underbrace{\begin{bmatrix} (0.25)^2 & 0 \\ 0 & (0.25)^2 \end{bmatrix} \varepsilon}_{\Sigma_y}, \quad (31)$$

with $\varepsilon \sim \mathcal{N}(0, I_2)$. We compare our approach with other ensemble methods that operate directly using the transition model and the observation operator to approximate the

filtering distribution at each step $k$ by an ensemble of particles. These methods are described in Appendix B. For each method and ensemble size, we perform 32 filtering runs and report the average skill, defined as the root mean square error (RMSE) between the ensemble mean and the true state $x^k$, on Figure 1.

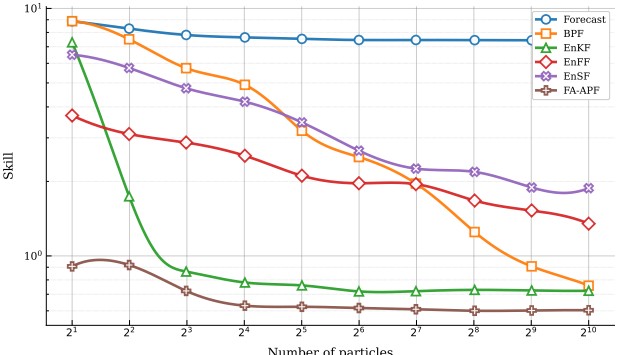

*Figure 1.* Evolution of the average skill as a function of the number of ensemble members.

Our method (brown curve on Figure 1) consistently yields lower errors than competing algorithms for a given number of members, and outperforms them even when using fewer members. In particular, FA-APF substantially outperforms the classical particle filter (BPF, orange curve on Figure 1), which is known to require many particles for reliable performance. This improvement is due to the use of the optimal proposal in FA-APF (whereas the classical BPF relies on the standard proposal $q(x^{k+1} \mid x^k, y^{k+1}) = p(x^{k+1} \mid x^k)$), and highlights the potential of particle filters for high-dimensional applications. We also note the strong performance of EnKF (Evensen, 2009) with a limited number of particles, explaining its widespread use in operational meteorology centers. Table 1 reports the results for all algorithms with a fixed ensemble size of $N = 256$. The first five rows refer to the previous experiment for which we observe the first and last components, while the last five rows correspond to a second experiment where we only observe the first component with a standard deviation of 0.25.

*Table 1.* Average skill, spread-to-skill ratio (SSR) and CRPS (see Appendix C) on 32 filtering runs for each method.

|  | SKILL ($\downarrow$) | SSR ($\approx 1$) | CRPS ($\downarrow$) |
|---|---|---|---|
| BPF | 1.25 | 0.32 | 2.88 |
| ENKF | 0.72 | 1.25 | 1.10 |
| ENSF | 2.18 | **0.78** | 4.23 |
| ENFF | 1.67 | 1.25 | 2.51 |
| FA-APF | **0.6** | 1.49 | **0.98** |
| BPF | 2.38 | **0.85** | 4.35 |
| ENKF | 2.79 | 1.21 | 4.35 |
| ENSF | 3.76 | 1.19 | 6.32 |
| ENFF | 3.67 | 1.39 | 5.96 |
| FA-APF | **1.87** | 1.25 | **3.05** |

## 4.2. Incompressible Navier-Stokes Flow

Another important aspect in practice is the robustness of the algorithm to sparse observations, that is, when the observation space has a much lower dimension than the state space. For example, methods such as EnSF (Bao et al., 2024) and EnFF (Transue et al., 2025), although theoretically elegant, perform significantly worse in such regimes (Si & Chen, 2025).

To evaluate our method under these conditions, we consider a high-dimensional system describing an incompressible fluid governed by the 2D Navier–Stokes equations with random forcing on the torus $\mathbb{T}^2 = [0, 2\pi]^2$

$$\mathrm{d}\omega + v \cdot \nabla\omega \, \mathrm{d}t = \nu\Delta\omega \, \mathrm{d}t - \alpha\omega \, \mathrm{d}t + \varepsilon\mathrm{d}\xi, \qquad (32)$$

where $\omega$ represents the vorticity, $v$ the velocity, and $\xi$ a white-in-time random forcing acting on Fourier modes. We directly rely on the dataset of Chen et al. (2024), which consists of $\mathcal{O}(10^5)$ consecutive vorticity snapshots sampled every 0.5 time units on a $256 \times 256$ grid, that we downsample to $128 \times 128$ for computational efficiency.

We use this dataset to train a denoiser by minimizing the loss function defined in Equation (14), using the same formalism and noise scheduler as the previous experiment on Lorenz'63. The backbone of the denoiser is a U-Net (Ronneberger et al., 2015) with $\mathcal{O}(10^7)$ parameters, and is trained for 20 epochs. At inference time, we use the same DDIM sampler with only 32 steps to accelerate generation when solving Equation (11).

For the experiments, we consider a subsampling operator that selects specific points of the grid, and a coarsening operator that pixelates system states (see Figure 2). Observations are corrupted with Gaussian noise of standard deviation 0.1. We consider observations at three dimensional levels, corresponding to the rows of Table 2. For instance, $(32, 32)$ indicates an observation with dimension $32 \times 32$, corresponding to $6.25\%$ of the full system state. Table 2 reports the average skill over ten filtering experiments for the coarsening operator (first three rows) and the subsampling operator (last three rows) for BPF, FA-APF, and FlowDAS (Chen et al., 2026) with 128 particles.

*Table 2.* Average skill for BPF, FlowDAS and FA-APF.

| $\mathcal{H}$ | $d$ | ALGORITHMS | | |
| | | BPF | FLOWDAS | FA-APF |
|---|---|---|---|---|
| COARSE | $(8, 8)$ | 3.02 | _2.78_ | **2.39** |
| | $(16, 16)$ | 2.88 | _0.74_ | **0.63** |
| | $(32, 32)$ | 2.80 | _0.19_ | **0.14** |
| SPARSE | $(8, 8)$ | _2.96_ | 3.08 | **2.30** |
| | $(16, 16)$ | 2.91 | _1.9_ | **1.12** |
| | $(32, 32)$ | 2.88 | _0.20_ | **0.13** |

We consistently observe a lower skill with FA-APF and strong qualitative results as in Figure 2, especially compared to the classical particle filter (BPF), which fails entirely under such high-dimensional settings, confirming that it is unusable in realistic geophysical systems (Van Leeuwen, 2015).

Our method is also more robust than FlowDAS, which can be considered as training-free if an interpolant of the dynamics is already available (Chen et al., 2024). Actually, as presented by Chen et al. (2026), FlowDAS corresponds to a degenerate fully adapted filter that propagates its single particle sequentially using the optimal proposal. Extending this approach to an ensemble of particles yields a pseudo FA-APF that ignores particle weights and does not select which particles to propagate at each step, resulting in a distribution that deviates from the true filtering distribution. Our algorithm can therefore be seen as a natural ensemble-based extension of FlowDAS, while remaining rigorously grounded in the particle filtering formalism.

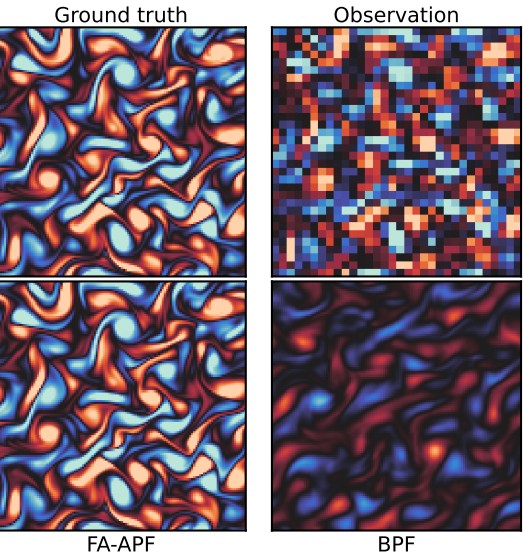

*Figure 2.* Ground truth, coarse $32 \times 32$ observation, FA-APF ensemble mean and BPF ensemble mean at the last step of a filtering experiment. Examples of trajectories are given in Appendix D.

Like FlowDAS, our method can be adapted to the stochastic interpolant framework (Albergo et al., 2025), as detailed in Appendix E. This framework corresponds to a family of generative models that generalize diffusion and flow matching (Lipman et al., 2023), and can thus be applied to a wide range of generative emulators. For clarity, we adopted the diffusion framework in this study, as it is the most established and widely used framework, and it was also the framework behind GenCast (Price et al., 2025), which we examined in the next experiment.

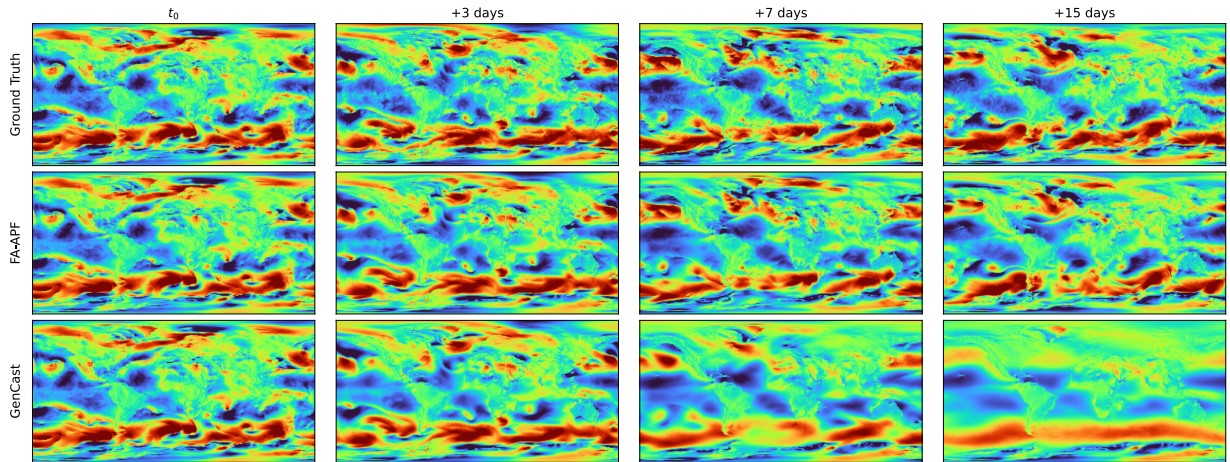

*Figure 3.* Comparison of the 10m U component of wind between the reference ERA5 trajectory (first row), the FA-APF ensemble mean obtained with realistic observations (second row), and the GenCast ensemble mean (third row) after 3, 7, and 15 days.

## 4.3. Medium-range weather forecasts (GenCast)

In the final experiment, we apply our method in a real-world scenario by leveraging the denoiser from GenCast (Price et al., 2025), a diffusion emulator of the atmosphere trained on the ERA5 reanalysis dataset (Hersbach et al., 2020). In this setting, the system state $x^k$ is high-dimensional, as it consists of 83 surface and atmospheric variables defined on a 1° latitude-longitude grid, resulting in $\mathcal{O}(10^6)$ variables.

An ensemble of $N = 256$ particles is initialized using the first state of a reference ERA5 trajectory that was not included in the training dataset. This trajectory is then used as ground truth to evaluate the metrics and to generate observations over a 15-day period, corresponding to 30 time steps with the 12-hour resolution of GenCast. We study two observation scenarios:

- **Sparse temperature observations**. We subsample and coarsen the latitude–longitude grid by retaining one point out of 4 in each direction and averaging $4 \times 4$ non-overlapping patches. We observe only temperature, the most readily available variable in practice from weather stations and satellites. This corresponds to observing approximately 1% of the full system state. Gaussian observation noise with standard deviation 0.2 is added to reflect modern temperature measurement accuracy.

- **Realistic observations**. We consider ground-based stations inspired by real-world station locations. These stations measure 4 of the 6 surface variables with Gaussian noise based on the performance of current measuring instruments. In addition, drawing inspiration from the setting of Andry et al. (2025), we include satellite observations of atmospheric temperature bands with Gaussian noise of 0.5K, reflecting the indirect nature of temperature retrieval from radiance measurements.

Figure 4 shows the evolution of the skill for two surface variables (U component of wind and temperature) across successive assimilation steps. Results are shown for our filtering method under both observation scenarios and for an ensemble of forecasts with the same ensemble size.

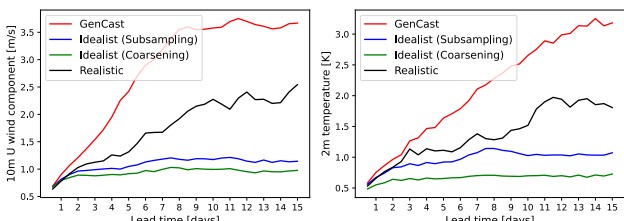

*Figure 4.* Skill comparison between FA-APF with sparse temperature observations (blue and green curves), FA-APF with realistic observations (black curve), and the ensemble of unconditional GenCast trajectories (red curve) for the surface variables.

For the first observation setting, based on subsampled and coarsened temperature observations, we obtain a stable approximation of the system state after 7 days of observations. This holds even for unobserved variables, such as wind in Figure 4, while maintaining a non-zero ensemble spread. Spread and additional skill curves are given in Appendix D.

Under the more realistic observation setting, convergence toward a stable skill across all variables is not achieved, even after 15 days of observations. This is mainly due to the strong spatial inhomogeneity of the observations, both at the surface and in the atmosphere, particularly near the poles, where errors grow rapidly in the absence of measurements. Nevertheless, this setting has intentionally been made challenging, and operational systems would typically have access to denser observations. Figure 3 illustrates the ensemble mean obtained by filtering in this setting, which remains qualitatively close to the true system state.

## 5. Related work

Variational methods such as 4D-Var (Le Dimet & Talagrand, 1986) have been widely used in operational centers (Brousseau et al., 2025) and are effective in practice. However, they rely on tangent linear and adjoint models, which are computationally expensive and may fail to capture strongly nonlinear dynamics, such as extreme weather events in the case of the atmosphere. Moreover, they provide only point estimates rather than full probabilistic predictions, preventing uncertainty quantification.

Ensemble Kalman filters (EnKF) and their variants, such as LETKF (Hunt et al., 2007), are another widely used class of methods in operational weather centers (Brousseau et al., 2025). They provide probabilistic state estimates at each assimilation step but are theoretically valid only under linear dynamics, linear observation operator, and Gaussian noise assumptions. In practice, heuristics such as covariance inflation and localization are needed to handle nonlinear, high-dimensional systems. Despite these limitations, our experiments show that EnKF variants remain competitive.

More recently, training-free methods based on generative models have been introduced (Bao et al., 2024; Transue et al., 2025). These methods rely only on the transition model $p(x^{k+1} \mid x^k)$ and the observation operator $\mathcal{H}$. Starting from the filtering distribution at time $k$, they forecast particles to the next time step and estimate a vector field that maps the current filtering distribution to the next predictive distribution. This vector field is then adjusted using gradients of the observation operator to converge toward the next filtering distribution. While elegant in theory, these approaches are effective only under dense observation regimes (Si & Chen, 2025), which are rare in practice.

Other generative approaches require training of specific models. The Score-Based Data Assimilation framework (Rozet & Louppe, 2023) focuses on Bayesian smoothing by training a local score network and combining these local scores to generate a full trajectory consistent with the observations. The DAISI framework (Andrae et al., 2025) proposes an iterative filtering algorithm based on a stochastic interpolant learned directly from the data. However, this method is not guaranteed to correspond to proper Bayesian filtering and depends on the efficiency of the forecast model.

Finally, our work is closely related to FlowDAS (Chen et al., 2026), which can be viewed as a training-free method adaptable to generative emulators of dynamical systems. However, FlowDAS is also not mathematically grounded, as its algorithm does not provide an exact solution to the Bayesian filtering problem, except in the trivial case of a single particle, thereby losing the advantage of ensemble methods for uncertainty quantification. Our method can thus be seen as a generalization of FlowDAS to proper Bayesian filtering.

## 6. Conclusion

In this work, we showed that generative emulators of dynamical systems can be adapted, without additional training, to address Bayesian filtering through an optimized version of particle filters (Petetin & Desbouvries, 2013). Although this variant is known to be more efficient than the classical particle filter (Slivinski & Snyder, 2016), it has remained largely impractical due to the difficulty of sampling from the optimal proposal, a limitation we address using training-free posterior sampling such as MMPS (Rozet et al., 2024).

Our results demonstrate that this approach consistently outperforms the classical particle filter and competing methods for a fixed ensemble size (Section 4.1), and remains effective in high-dimensional settings with sparse observations (Sections 4.2 and 4.3), successfully scaling to problems with up to $\mathcal{O}(10^6)$ variables. These findings show that our method can be successfully applied to realistic large-scale problems without relying on linearization or restrictive assumptions on the system dynamics.

## 7. Limitations & Future work

A fundamental limitation of our approach is its reliance on generative emulators of the system dynamics. While such models are becoming increasingly popular (Rozet et al., 2025; Larsson et al., 2025; Finn et al., 2024; Price et al., 2025), in particular because they preserve uncertainty and avoid long-term over-smoothing, they are still less used than classical numerical solvers and deterministic neural emulators (Barros et al., 1995; Lam et al., 2023).

Sampling from the optimal proposal is also computationally expensive, as it requires differentiating the denoiser at each step of the reverse diffusion process to estimate the score of the posterior. For very large-scale systems such as GenCast (Section 4.3), this results in significant memory overhead (see Appendix F for further details on the complexity of Algorithm 1). An important direction for future work is to develop training-free posterior sampling methods that remain accurate for highly nonlinear observation operators while being computationally efficient.

The performance of our method is further constrained by the quality of the transition model, the approximations used to sample from the optimal proposal and compute particle weights, and the use of inflation in Algorithm 1 to mitigate weight degeneracy. Additional details on these approximations are given in Appendix F.

Finally, assuming access to generative models capable of directly sampling joint distributions $p(x^k, x^{k+1})$, a natural extension of this work would be to adapt the proposed framework for training-free Bayesian smoothing (Doucet & Johansen, 2009; Klaas et al., 2006).

## Acknowledgments and Disclosure of Funding

We acknowledge the support of the F.R.S.-FNRS (Belgium) and its funding of the Mosaic project (MIS F.4536.25). François Rozet is a research fellow of the F.R.S.-FNRS and acknowledges its financial support.

The present research benefited from computational resources made available on Lucia, the Tier-1 supercomputer of the Walloon Region, infrastructure funded by the Walloon Region under the grant agreement n°1910247. We also acknowledge the support of NVIDIA Corporation for computing resources offered through the Academic Grant Program.

## Impact Statement

This paper presents work whose goal is to advance the fields of Machine Learning and Data Assimilation. There are many potential societal consequences of our work, none of which we feel must be specifically highlighted here.

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

## A. Tweedie's formulas

**Theorem A.1.** *Assuming that $p_t(x_t \mid x) = \mathcal{N}(x_t \mid \alpha_t x, \Sigma_t)$, the first and second moments of $p_t(x \mid x_t)$ are linked to the score function $\nabla_{x_t} \log p_t(x_t)$ used in Equation* (11) *through*

$$\mathbb{E}[x \mid x_t] = \alpha_t^{-1} \left[ x_t + \Sigma_t \nabla_{x_t} \log p_t(x_t) \right], \tag{33}$$

$$\mathbb{V}[x \mid x_t] = \alpha_t^{-2} \left[ \Sigma_t + \Sigma_t \nabla_{x_t}^2 \log p_t(x_t) \right]. \tag{34}$$

We provide proofs of this theorem for completeness, even though it is a well known result (Efron, 2011).

*Proof.*

$$\nabla_{x_t} \log p_t(x_t) = \frac{1}{p_t(x_t)} \int \nabla_{x_t} p_t(x_t, x) \mathrm{d}x$$

$$= \frac{1}{p_t(x_t)} \int p_t(x_t, x) \nabla_{x_t} \log p_t(x_t, x) \mathrm{d}x$$

$$= \int p_t(x \mid x_t) \nabla_{x_t} \log p_t(x_t \mid x) \mathrm{d}x$$

$$= \int p_t(x \mid x_t) \Sigma_t^{-1} (\alpha_t x - x_t) \, \mathrm{d}x$$

$$= \Sigma_t^{-1} (\alpha_t \mathbb{E}[x \mid x_t] - x_t)$$

$\square$

*Proof.*

$$\nabla_{x_t}^2 \log p_t(x_t) = \nabla_{x_t} \left[ \nabla_{x_t}^\top \log p_t(x_t) \right]$$

$$= \alpha_t \nabla_{x_t}^\top \mathbb{E}[x \mid x_t] \Sigma_t^{-1} - \Sigma_t^{-1}$$

$$= \alpha_t \left[ \int \nabla_{x_t} p_t(x \mid x_t) x^\top \mathrm{d}x \right] \Sigma_t^{-1} - \Sigma_t^{-1}$$

$$= \alpha_t \left[ \int p_t(x \mid x_t) \nabla_{x_t} \log p_t(x \mid x_t) x^\top \mathrm{d}x \right] \Sigma_t^{-1} - \Sigma_t^{-1}$$

$$= \alpha_t \left[ \int p_t(x \mid x_t) \alpha_t \Sigma_t^{-1} (x - \mathbb{E}[x \mid x_t]) x^\top \mathrm{d}x \right] \Sigma_t^{-1} - \Sigma_t^{-1}$$

$$= \alpha_t^2 \Sigma_t^{-1} \left( \mathbb{E}[xx^\top \mid x_t] - \mathbb{E}[x \mid x_t] \mathbb{E}[x \mid x_t]^\top \right) \Sigma_t^{-1} - \Sigma_t^{-1}$$

$$= \alpha_t^2 \Sigma_t^{-1} \mathbb{V}[x \mid x_t] \Sigma_t^{-1} - \Sigma_t^{-1}$$

$\square$

# B. Training-free ensemble methods

In the Lorenz'63 experiment, we compared our method with four other training-free ensemble algorithms. These methods approximate the Bayesian filtering distribution at each assimilation step $k$ using an ensemble of $N$ particles, relying only on the transition distribution $p(x^{k+1} \mid x^k)$ and the observation operator $\mathcal{H}$.

## B.1. Bootstrap Particle Filter (BPF)(van Leeuwen et al., 2019)

The Bootstrap Particle Filter (BPF, Algorithm 3) is the simplest particle filter that samples new particles from the transition distribution $p(x^{k+1} \mid x^k)$. In theory, it converges to the Bayesian filtering distribution, even with nonlinear dynamics and observation operators, but requires a large number of particles to perform well.

---

**Algorithm 3** Bootstrap Particle Filter (BPF)

---

1: **Inputs:** $p(x^0)$, $N$, $K$, $N_{\text{thr}}$
2: $x_i^0 \sim p(x^0)$
3: $w_i^0 \leftarrow 1/N$
4: **for** $k = 0$ **to** $K - 1$ **do**
5:     $x_i^{k+1} \sim p(x^{k+1} \mid x_i^k)$
6:     $\hat{w}_i^{k+1} \leftarrow p(y^{k+1} \mid x_i^{k+1})$
7:     $w_i^{k+1} \leftarrow \hat{w}_i^{k+1} / \sum_{j=1}^N \hat{w}_j^{k+1}$
8:     $N_{\text{eff}} \leftarrow 1 / \sum_{i=1}^N (w_i^{k+1})^2$
9:     **if** $N_{\text{eff}} < N_{\text{thr}}$ **then**
10:         do resampling
11:     **end if**
12: **end for**
13: **Return** $\mu_x^k = \sum_{i=1}^N w_i^k \delta_{x_i^k}$ for all $k \in [1, K]$

---

## B.2. Ensemble Kalman Filter (EnKF) (Evensen, 2009)

The Ensemble Kalman Filter (EnKF, Algorithm 4) is a popular ensemble-based method for Bayesian filtering. Although it assumes near-Gaussian distributions, EnKF performs well in practice and its variants are widely used in high-dimensional applications.

---

**Algorithm 4** Ensemble Kalman Filter (EnKF)

---

1: **Inputs:** $p(x^0)$, $N$, $K$
2: $x_i^0 \sim p(x^0)$
3: **for** $k = 0$ **to** $K - 1$ **do**
4:     $x_i^f \sim p(x^{k+1} \mid x_i^k)$
5:     $h_i^f \leftarrow \mathcal{H}(x_i^f)$
6:     $\bar{x}^f \leftarrow \frac{1}{N} \sum_{i=1}^N x_i^f$
7:     $\bar{h}^f \leftarrow \frac{1}{N} \sum_{i=1}^N h_i^f$
8:     $P_{yy} \leftarrow \frac{1}{N-1} \sum_{i=1}^N (h_i^f - \bar{h}^f)(h_i^f - \bar{h}^f)^T$
9:     $P_{xy} \leftarrow \frac{1}{N-1} \sum_{i=1}^N (x_i^f - \bar{x}^f)(h_i^f - \bar{h}^f)^T$
10:     $G \leftarrow P_{xy}(P_{yy} + \Sigma_y)^{-1}$
11:     **for** $i = 1$ **to** $N$ **do**
12:         $\epsilon_i \sim \mathcal{N}(0, \Sigma_y)$
13:         $d_i \leftarrow y^{k+1} + \epsilon_i$
14:         $x_i^{k+1} \leftarrow x_i^f + G(d_i - h_i^f)$
15:     **end for**
16: **end for**
17: **Return** $\mu_x^k = \frac{1}{N} \sum_{i=1}^N \delta_{x_i^k}$ for all $k \in [1, K]$

---

### B.3. Ensemble Score Filter (EnSF) (Bao et al., 2024)

The Ensemble Score Filter (EnSF, Algorithm 5) is a recent Bayesian filtering algorithm inspired by diffusion models. The method consists in estimating the score by Monte Carlo and then updating it using the gradient of the likelihood.

---

**Algorithm 5** Ensemble Score Filter (EnSF)

---

1: **Inputs:** $p(x^0)$, $N$, $\Delta_t$, $K$, $\alpha(\cdot)$, $\beta(\cdot)$, $b(\cdot)$, $\sigma(\cdot)$
2: $x_i^0 \sim p(x^0)$
3: **for** $k = 0$ **to** $K - 1$ **do**
4:    $x_i^f \sim p(x^{k+1} \mid x_i^k)$
5:    **for** $i = 1$ **to** $N$ **do**
6:       $z_i \sim \mathcal{N}(0, \beta(1)^2 I)$
7:       **for** $t$ in $[1, 1 - \Delta_t, \cdots, \Delta_t]$ **do**
8:          $w_j \leftarrow \mathcal{N}(z_i \mid \alpha(t)x_j^f, \beta_t^2 I) / \sum_{k=1}^N \left[ \mathcal{N}(z_i \mid \alpha(t)x_k^f, \beta_t^2 I) \right]$
9:          $s_x \leftarrow -\sum_{j=1}^N \left( \frac{z_i - \alpha(t)x_j^f}{\beta(t)^2} \right) w_j$
10:         $s_{x,y} \leftarrow s_x + (1 - t) \times \nabla_{z_i} \log p(y^{k+1} \mid z_i)$
11:         $z_i \leftarrow z_i - \left[ b(t)z_i - \sigma(t)^2 s_{x,y} \right] \Delta_t - \sigma(t)\sqrt{\Delta_t}\varepsilon, \ \varepsilon \sim \mathcal{N}(0, I)$
12:       **end for**
13:       $x_i^{k+1} \leftarrow z_i$
14:    **end for**
15: **end for**
16: **Return** $\mu_x^k = \frac{1}{N} \sum_{i=1}^N \delta_{x_i^k}$ for all $k \in [1, K]$

---

### B.4. Ensemble Flow Filter (EnFF) (Transue et al., 2025)

The Ensemble Flow Filter (EnFF, Algorithm 6) is a flow-matching-based variant of the EnSF algorithm presented above. The idea is to take advantage of the flow matching framework to construct straighter paths between two successive filtering distributions, leading to more efficient sampling.

---

**Algorithm 6** Ensemble Flow Filter (EnFF)

---

1: **Inputs:** $p(x^0)$, $N$, $K$, $\Delta_t$, $\lambda$, $\sigma_{\min}$
2: $x_i^0 \sim p(x^0)$
3: **for** $k = 0$ **to** $K - 1$ **do**
4:    $x_i^f \sim p(x^{k+1} \mid x_i^k)$
5:    **for** $i = 1$ **to** $N$ **do**
6:       $z_i \leftarrow x_i^k$
7:       **for** $t$ in $[0, \Delta_t, \cdots, 1 - \Delta_t]$ **do**
8:          $w_j \leftarrow \mathcal{N}(z_i \mid tx_j^f + (1 - t)x_k^k, \sigma_{\min}^2 I) / \sum_{k=1}^N \left[ \mathcal{N}(z_i \mid tx_k^f + (1 - t)x_k^k, \sigma_{\min}^2 I) \right]$
9:          $u \leftarrow \sum_{j=1}^N \left( x_j^f - x_j^k \right) w_j$
10:         $\hat{z}_i \leftarrow z_i + (1 - t) \times u$
11:         $\tilde{u} \leftarrow u - \lambda \times \nabla_{\hat{z}_i} \log p(y^{k+1} \mid \hat{z}_i)$
12:         $z_i \leftarrow z_i + \Delta_t \times \tilde{u}$
13:       **end for**
14:       $x_i^{k+1} \leftarrow z_i$
15:    **end for**
16: **end for**
17: **Return** $\mu_x^k = \frac{1}{N} \sum_{i=1}^N \delta_{x_i^k}$ for all $k \in [1, K]$

---

## C. Metrics

### C.1. Skill

The skill of an ensemble of $N$ particles $\left\{x_i^k\right\}_{1 \leq i \leq N}$ at time $k$ is defined as the RMSE of the ensemble mean

$$\text{Skill} = \sqrt{\left\langle \left( u^k - \frac{1}{N} \sum_{i=1}^{N} x_i^k \right)^2 \right\rangle} \tag{35}$$

where $\langle \cdot \rangle$ denotes the spatial mean operator and $u^k$ the ground truth at step $k$. In Section 4.1 and 4.2, we compute the skill for each assimilation steps and then compute the average skill over the experiment.

### C.2. Spread

The spread of an ensemble of $N$ particles $\left\{x_i^k\right\}_{1 \leq i \leq N}$ at time $k$ is defined as the ensemble standard deviation

$$\text{Spread} = \sqrt{\left\langle \frac{1}{N-1} \sum_{i=1}^{N} \left( x_i^k - \frac{1}{N} \sum_{j=1}^{N} x_j^k \right)^2 \right\rangle}. \tag{36}$$

### C.3. Spread-to-skill ratio (SSR)

As shown by Fortin et al. (2014), a well-calibrated forecast should have a spread-to-skill ratio of 1, which is a necessary but not sufficient condition. Ratios below one indicate overconfident estimates, whereas ratios above one indicate underconfident estimates. In Section 4.1, we compute the SSR for each assimilation steps and then compute the average SSR over the experiment.

### C.4. Continuous ranked probability score (CRPS)

The CRPS score (Gneiting & Raftery, 2007) of an assimilation experiment is defined as

$$\text{CRPS} = \frac{1}{K} \sum_{k=1}^{K} \left( \frac{1}{N} \sum_{i=1}^{N} \|u^k - x_i^k\|_{L_1} - \frac{1}{2N(N-1)} \sum_{i=1}^{N} \sum_{j=1}^{N} \|x_i^k - x_j^k\|_{L_1} \right) \tag{37}$$

where $K$ corresponds to the number of assimilation steps, $N$ to the number of particles, $\left\{x_i^k\right\}_{1 \leq i \leq N}$ to the ensemble of particles at time $k$, and $u^k$ to the ground truth at time $k$. The first term penalizes the average divergence from the ground truth while the second term encourages spread. Therefore, the CRPS is lowest when the distribution of the ensemble matches the ground-truth distribution.

# D. Additional results

## D.1. Incompressible Navier-Stokes Flow

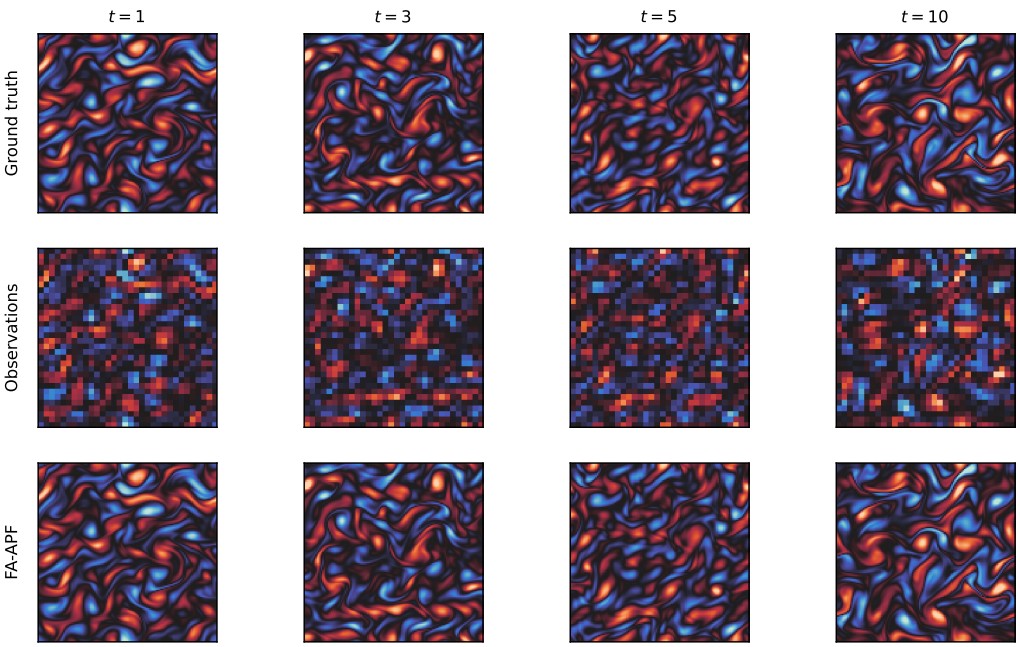

*Figure 5.* Ground truth, $32 \times 32$ coarse observation, and FA-APF ensemble mean at different time steps during a filtering experiment.

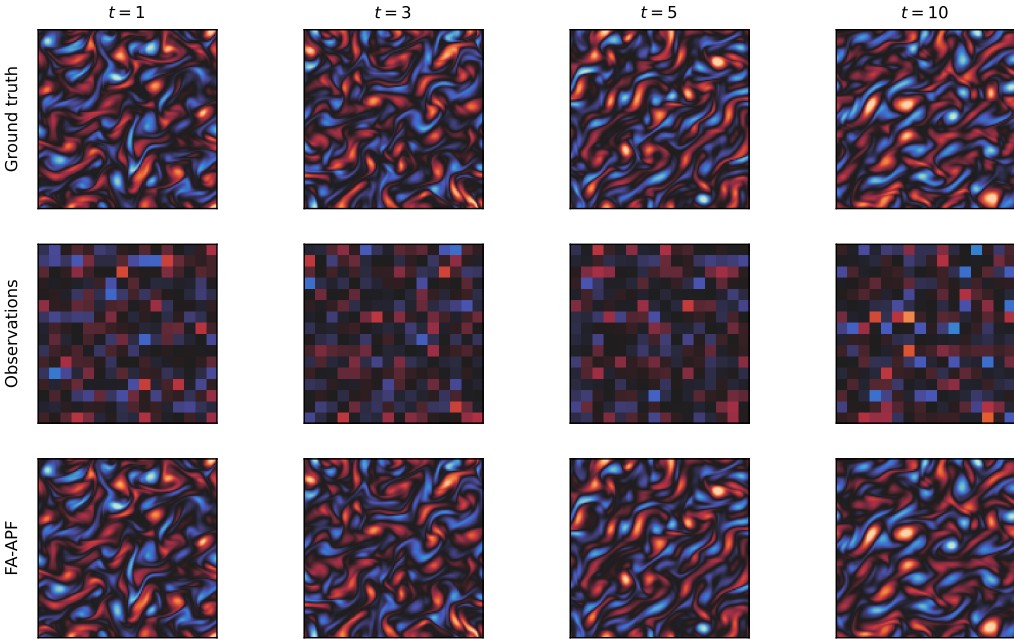

*Figure 6.* Ground truth, $16 \times 16$ coarse observation, and FA-APF ensemble mean at different time steps during a filtering experiment.

## D.2. Medium-range weather forecasts (GenCast)

### D.2.1. METRICS FOR ALL VARIABLES

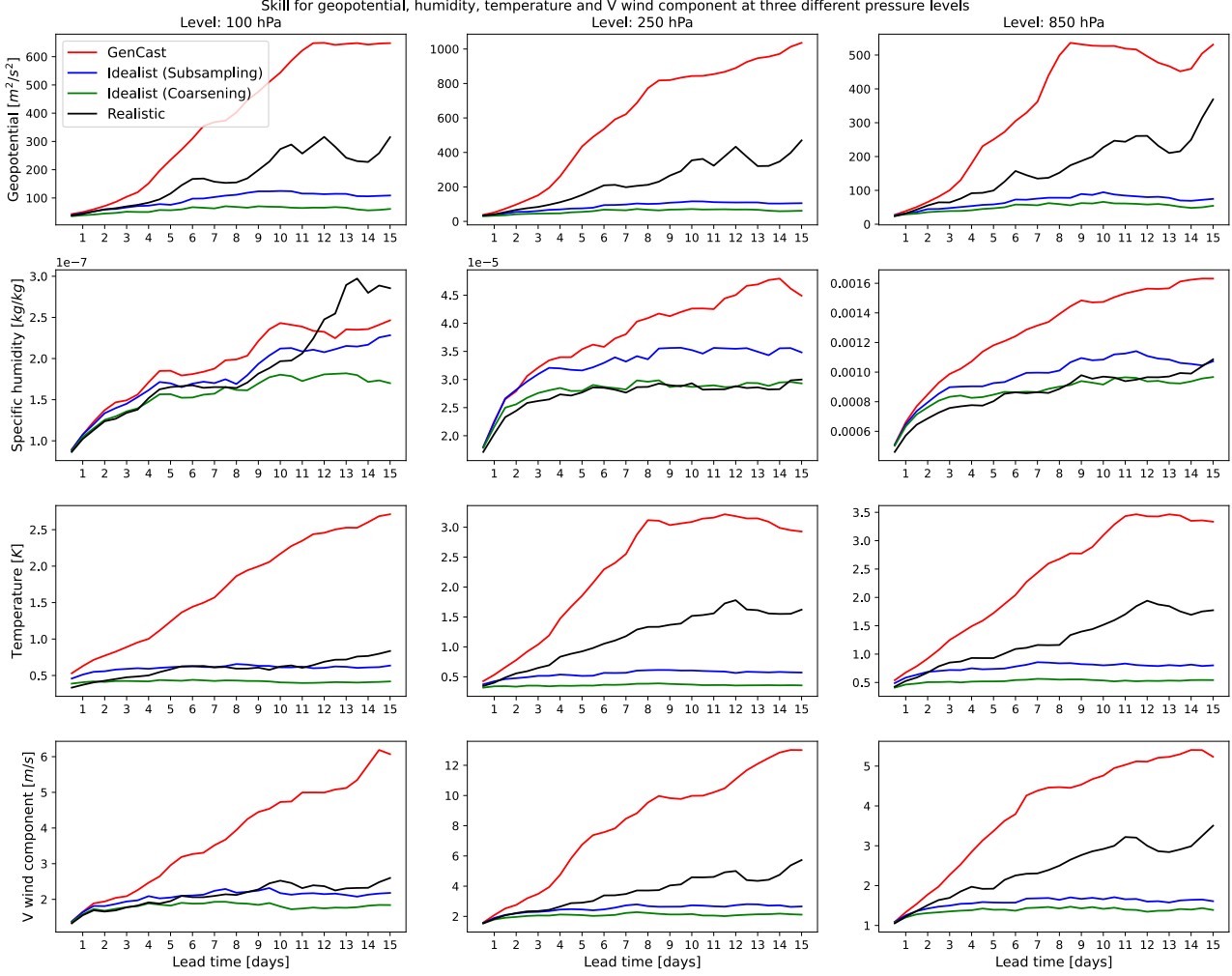

*Figure 7.* Skill for temperature, geopotential, V component of wind and specific humidity at three different pressure levels (100, 250 and 850 hPa). For experiments with sparse temperature observations (blue and green curves), the skill reaches a plateau after a certain number of time steps for all variables (even those that are not observed), well below the one of GenCast's forecasts.

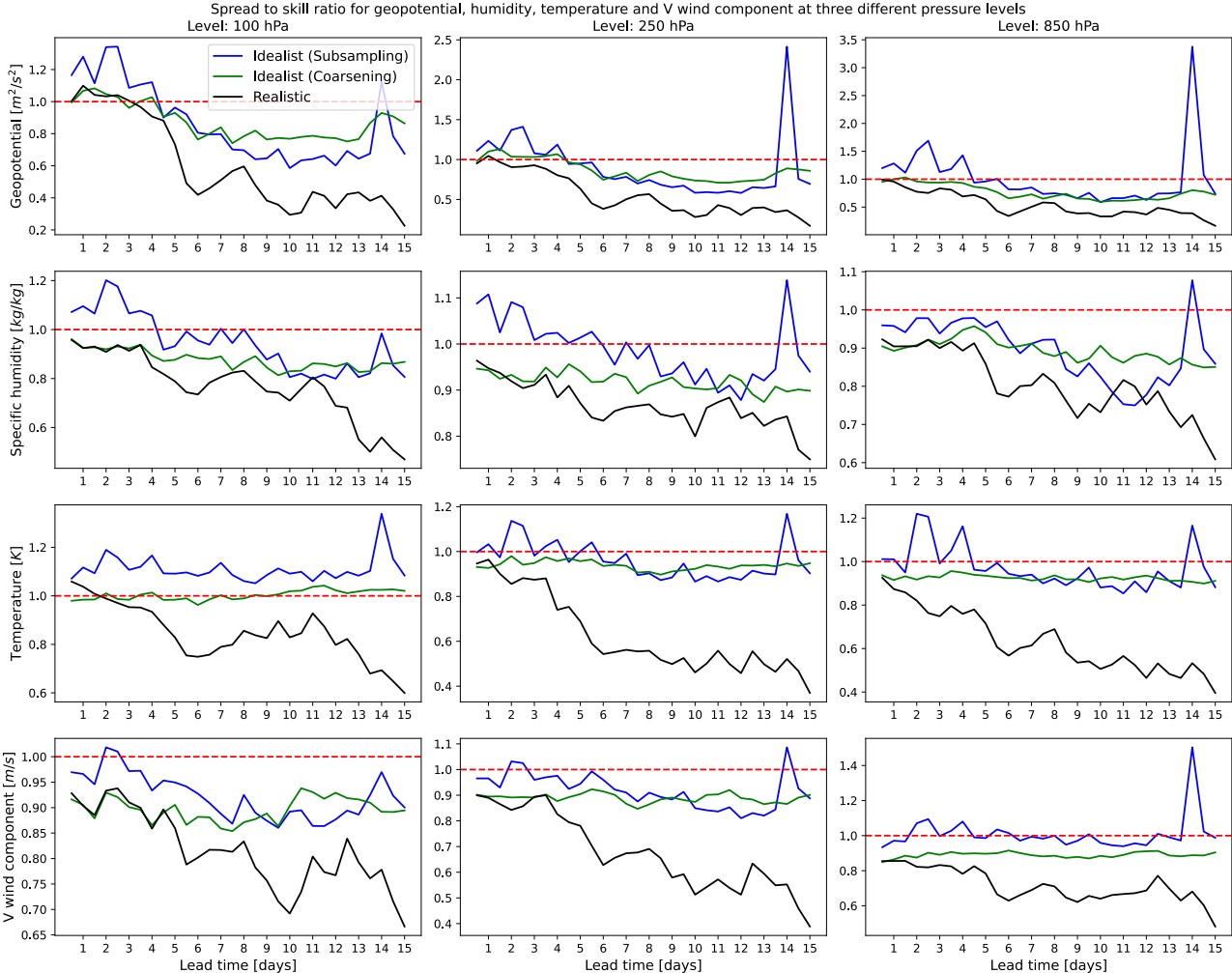

*Figure 8.* Spread-to-skill ratio for temperature, geopotential, V component of wind and specific humidity at three different pressure levels (100, 250 and 850 hPa). For experiments with sparse temperature observations (blue and green curves), the ratio is close to 1, indicating that ensembles are well calibrated.

D.2.2. VISUALIZATION OF TRAJECTORIES

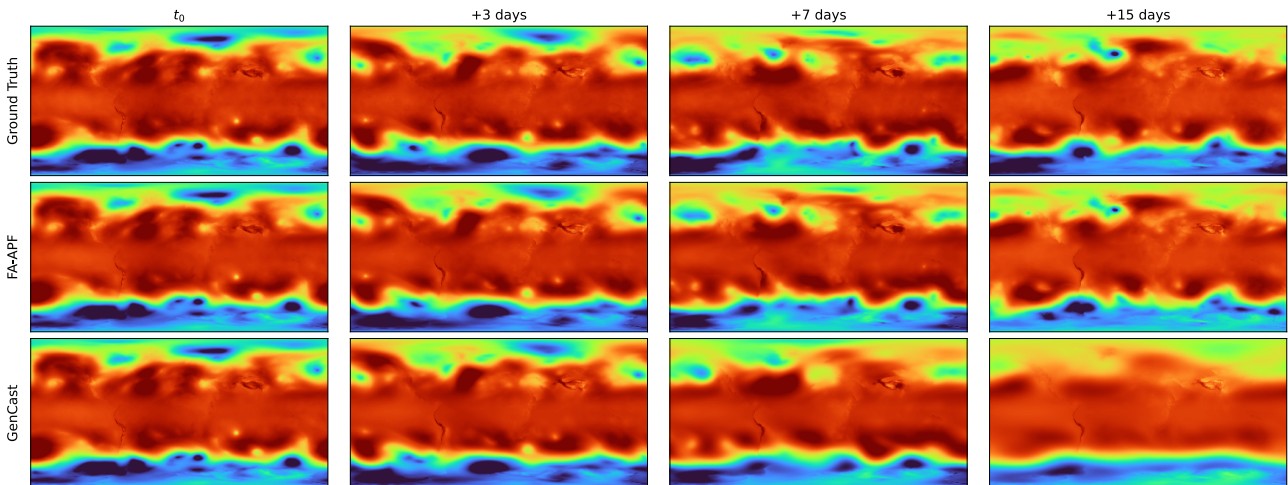

*Figure 9.* Comparison of the geopotential at 500 hPa between the reference ERA5 trajectory (first row), the FA-APF ensemble mean with realistic observations (second row), and the GenCast ensemble mean (third row) after 3, 7, and 15 days. The ensemble mean of FA-APF remains qualitatively close to the ground truth, even under difficult observation conditions.

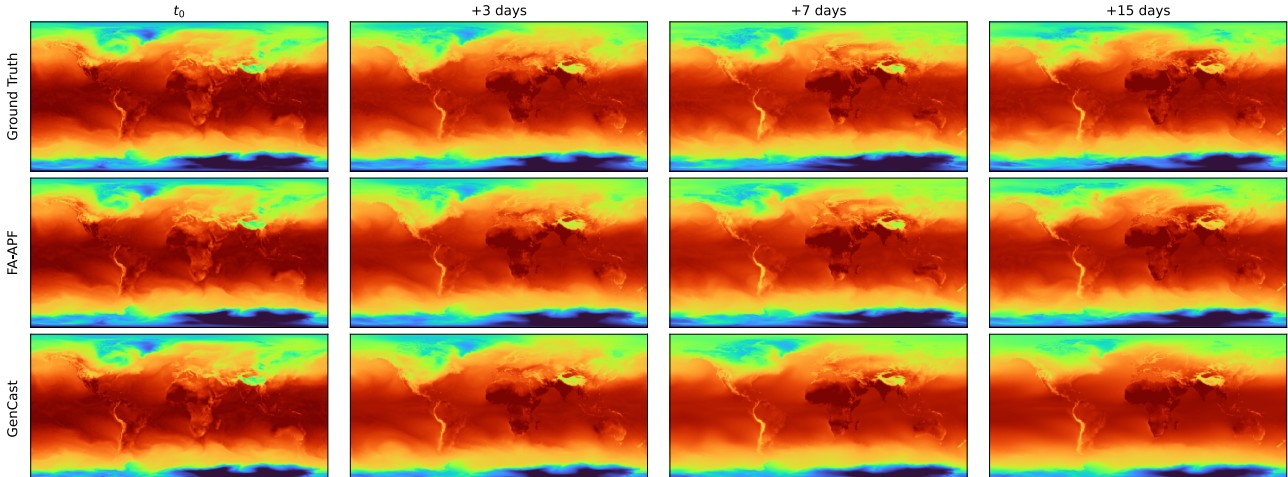

*Figure 10.* Comparison of surface temperature between the reference ERA5 trajectory (first row), the FA-APF ensemble mean with realistic observations (second row), and the GenCast ensemble mean (third row) after 3, 7, and 15 days. The ensemble mean of FA-APF remains qualitatively close to the ground truth, even under difficult observation conditions.

### D.2.3. POSTERIOR PREDICTIVE CHECK (PPC)

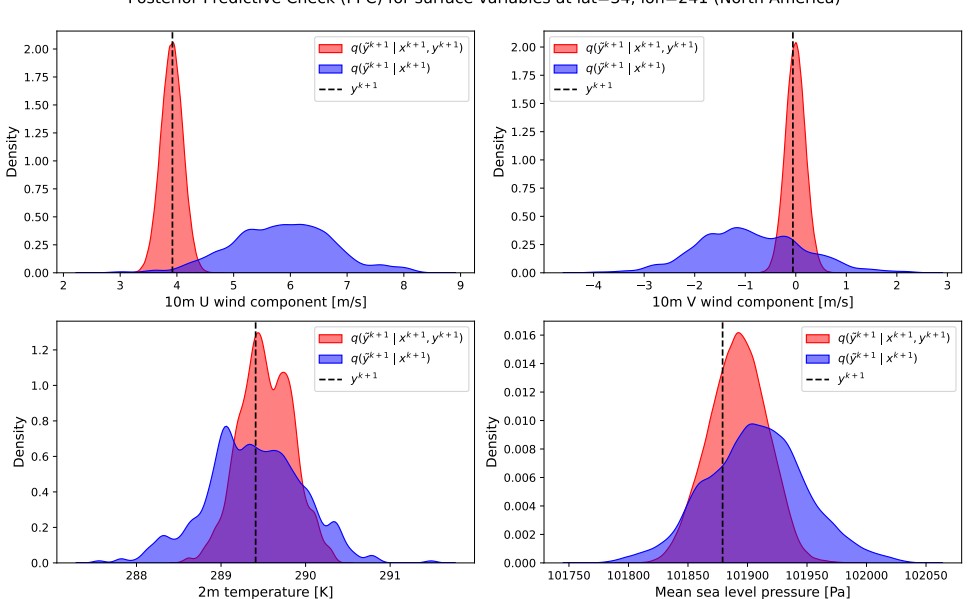

*Figure 11.* Comparison between the distributions of conditional samples (red curve, generated using the optimal proposal) and unconditional samples (blue curve, generated with GenCast without conditioning) at an observed grid point (in North America). Observations (black dotted lines) are more likely in the distribution of conditional samples.

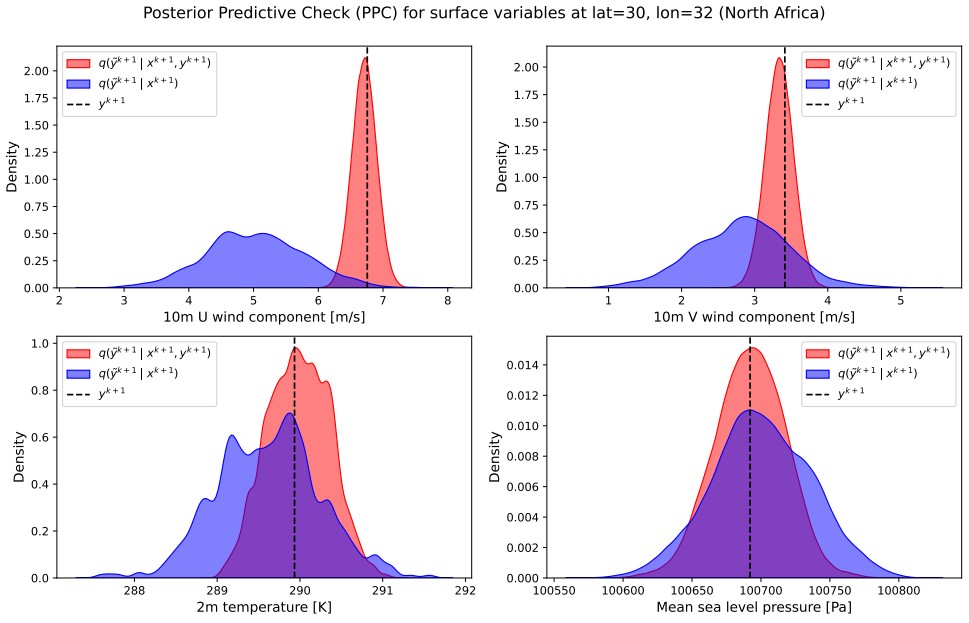

*Figure 12.* Comparison between the distributions of conditional samples (red curve, generated using the optimal proposal) and unconditional samples (blue curve, generated with GenCast without conditioning) at an unobserved grid point (in North Africa). Observations (black dotted lines) are more likely in the distribution of conditional samples.

# E. Extension to stochastic interpolants

## E.1. Stochastic interpolants for probabilistic forecasting

Stochastic interpolants (Albergo et al., 2025) are a class of generative models that generalize diffusion (Song et al., 2021b) and flow matching (Lipman et al., 2023). They are designed to learn a transport between an easily sampled distribution $\rho_0$ and a target distribution $\rho_1$. In their linear form, commonly used in practice, they are defined by

$$x_t = \alpha_t x_0 + \beta_t x_1 + \gamma_t z, \ t \in [0, 1], \tag{38}$$

where $(x_0, x_1)$ is a data pair drawn from a joint measure $\nu(\mathrm{d}x_0, \mathrm{d}x_1)$ with marginals $\rho_0(\mathrm{d}x_0)$ and $\rho_1(\mathrm{d}x_1)$, $z \sim \mathcal{N}(0, I)$ with $(x_0, x_1) \perp z$, and $\alpha, \beta, \gamma$ continuous functions on $[0, 1]$ that satisfy the following boundary conditions

$$\alpha_0 = \beta_1 = 1; \alpha_1 = \beta_0 = 0; \gamma_0 = \gamma_1 = 0. \tag{39}$$

As shown by Chen et al., stochastic interpolants can be adapted for probabilistic forecasting. Given a dataset of successive states $\{(x^k, x^{k+1})_i\}_{i \in \mathcal{I}}$ from a dynamical system, one may set $x_1 = x^{k+1}$ and train a neural network $b_\theta$ to learn the velocity given $x^k$. The network is trained by minimizing

$$\mathcal{L}(\theta) = \int_0^1 \mathbb{E}\left[\left\|\dot{x}_t - b_\theta(t, x_t, x^k)\right\|_2^2\right] \mathrm{d}t, \tag{40}$$

whose theoretical minimizer is $b_t(x_t, x^k) = \mathbb{E}[\dot{x}_t \mid x_t, x^k]$. Sampling from the transition law $p(x^{k+1} \mid x^k)$ is then performed by solving the forward generative equation

$$dx_t = b_t^F(x_t, x^k)dt + \sqrt{2\varepsilon(t)}dw_t, \tag{41}$$

$$b_t^F(x_t, x^k) = b_t(x_t, x^k) + \varepsilon(t)s_t^x(x_t, x^k), \tag{42}$$

from $x_0 \sim \rho_0$, where $s_t^x(x_t, x^k)$ denotes the score of the conditional density of $x_t$ given $x^k$, and $\varepsilon(t) \geq 0$ is a diffusion coefficient. The score $s_t^x(x_t, x^k)$ is not known a priori but can be expressed in terms of the velocity. For instance, when $x_0 = x^k$ the relation between score and velocity reads

$$s_t^x(x_t, x^k) = \frac{(\alpha_t \dot{\beta}_t - \dot{\alpha}_t \beta_t)x^k - \dot{\beta}_t x_t + \beta_t b_t(x_t, x^k)}{\gamma_t(\dot{\beta}_t \gamma_t - \beta_t \dot{\gamma}_t)}. \tag{43}$$

Using Equations (41) and (43), we can then generate probable future states $x^{k+1}$ from a current state $x^k$, and thus emulate the system dynamics in an autoregressive manner.

## E.2. Sampling from the optimal proposal with stochastic interpolants

To apply the FA-APF, we must sample from the optimal proposal distribution $p(x^{k+1} \mid x^k, y^{k+1})$, which is generally infeasible for standard simulators. However, as shown by Chen et al. and Andrae et al., this distribution can be accessed using stochastic interpolants. The key idea is to incorporate the observation $y^{k+1}$ into the generative dynamics defined in Equation (41). This leads to the following posterior forward equation

$$dx_t = b_t^F(x_t, x^k, y^{k+1})dt + \sqrt{2\varepsilon(t)}dw_t. \tag{44}$$

The forward drift $b_t^F(x_t, x^k, y^{k+1})$ is defined as the sum of a conditional velocity and a posterior score term

$$b_t(x_t, x^k, y^{k+1}) + \varepsilon(t)s_t^{x,y}(x_t, x^k, y^{k+1}), \tag{45}$$

where $s_t^{x,y}(x_t, x^k, y^{k+1})$ is the score of the density of $x_t$ given $x^k$ and $y^{k+1}$. The conditional velocity is not known a priori but, once again, can be derived directly from the velocity

$$b_t(x_t, x^k, y^{k+1}) = b_t(x_t, x^k) + \lambda_t s_t^y(x_t, x^k, y^{k+1}), \tag{46}$$

where $s_t^y(x_t, x^k, y^{k+1})$ is the score of the likelihood and $\lambda_t$ is a time-dependent coefficient given by

$$\lambda_t = \frac{\gamma_t(\dot{\beta}_t \gamma_t - \beta_t \dot{\gamma}_t)}{\beta_t}. \tag{47}$$

Thanks to Bayes' rule, the posterior score in Equation (45) can be decomposed as

$$s_t^{x,y}(x_t, x^k, y^{k+1}) = s_t^x(x_t, x^k) + s_t^y(x_t, x^k, y^{k+1}). \tag{48}$$

Since the prior score is already available from the velocity through Equation (43), the only unknown quantity that remains to be computed is the likelihood score $s_t^y(x_t, x^k, y^{k+1})$. To do so, we can use MMPS, the method introduced in Section 3.2.

Putting all these elements together, the forward drift conditioned on an observation can be fully computed from the learned interpolant without additional training. The resulting procedure is summarized in Algorithm 7.

---

**Algorithm 7** Computation of $b_t^F(x_t, x^k, y^{k+1})$

---

**Inputs:** $t$, $x_t$, $x^k$, $b_\theta$, $y^{k+1}$, $\varepsilon(\cdot)$

$b_x \leftarrow b_\theta(t, x_t, x^k)$

$s_x \leftarrow \dfrac{(\alpha_t \dot{\beta}_t - \dot{\alpha}_t \beta_t) x^k - \dot{\beta}_t x_t + \beta_t b_x}{\gamma_t(\dot{\beta}_t \gamma_t - \beta_t \dot{\gamma}_t)}$ (Eq. 43)

$s_y \leftarrow \mathrm{MMPS}(t, x_t, x^k, y^{k+1})$ (Eq. 20)

$b_{x,y} \leftarrow b_x + \dfrac{\gamma_t(\dot{\beta}_t \gamma_t - \beta_t \dot{\gamma}_t)}{\beta_t} s_y$ (Eq. 46)

$s_{x,y} \leftarrow s_x + s_y$ (Eq. 48)

$b^F = b_{x,y} + \varepsilon(t) s_{x,y}$ (Eq. 45)

**Return** $b^F$

---

### E.3. Computing weights with stochastic interpolants

The other important ingredient required to apply the FA-APF is the computation of particle weights (lines 5–10 of Algorithm 1). To do so, we can use the same approximation as in Section 3.3. It requires calculating $\mathbb{E}[x^{k+1} \mid x_i^k]$, which can be done directly using the learned velocity

$$\mathbb{E}[x^{k+1} \mid x_i^k] = \frac{b_0(x_i^k, x_i^k) - \dot{\alpha}_0 x_i^k}{\dot{\beta}_0}, \tag{49}$$

under the assumption that $\dot{\beta}_0 \neq 0$.

# F. Additional details on Algorithm 1

### F.1. Computational complexity

Using the notation of Algorithm 1, let $K$ denote the number of filtering steps, $N$ the number of particles, $T$ the number of diffusion steps required to solve Eq. (11), and $C_{\text{step}}$ the cost of a single diffusion step. Since the computational cost is dominated by posterior sampling (line 13 of Algorithm 1), the overall complexity of the algorithm is $\mathcal{O}\left(K \times N \times T \times C_{\text{step}}\right)$. However, because posterior sampling is fully parallelizable across particles, the effective complexity can be reduced by a factor $N$, and we therefore omit the explicit loop over particles in Algorithm 1 for readability.

As explained in Section 3, each diffusion step is based on MMPS (Rozet et al., 2024) and therefore involves solving a linear system (see Eq. 20). In practice, this system is solved iteratively using GMRES (Saad & Schultz, 1986), which involves vector–Jacobian products and leads to a cost of approximately $C_{\text{step}} = \mathcal{O}(M \times C_{\text{denoiser}})$, where $M$ is the number of GMRES iterations and $C_{\text{denoiser}}$ the cost of a pass through the denoiser. Thus, if posterior sampling can be parallelized across particles, the computational cost of the proposed method is $\mathcal{O}(K \times T \times M \times C_{\text{denoiser}})$. Importantly, FA-APF is compatible with any posterior sampling methods, including computationally cheaper alternatives such as DPS (Chung et al., 2023), although we adopt MMPS here due to its superior empirical performance.

Regarding memory complexity, the main bottleneck also comes from solving the linear system at each diffusion step when sampling from the optimal proposal. Indeed, since the covariance matrix $\mathbb{V}[x^{k+1} \mid x_t^{k+1}, x^k]$ is too large to be stored explicitly in high-dimensional systems, we instead solve a linear system using implicit access to $\mathbb{V}[x^{k+1} \mid x_t^{k+1}, x^k]$ through the second-order Tweedie's formula (see Eq. 21). This requires the Jacobian of the denoiser, which is obtained via automatic differentiation at the cost of approximately storing the denoiser activations. For large models such as GenCast (Section 4.3), this can be particularly demanding in terms of VRAM. However, we were able to run the 1° resolution denoiser on H100 GPUs with 80 Gb of VRAM without relying on memory-saving techniques such as gradient checkpointing.

### F.2. Approximations

Several approximations are introduced in Algorithm 1 to make FA-APF tractable in practice. First, the posterior sampling method (MMPS, Rozet et al.) used to sample from the optimal proposal is not exact. Indeed, the local diffusion distribution $p(x^{k+1} \mid x_t^{k+1}, x^k)$ is approximated by a Gaussian distribution whose moments are estimated using Tweedie's formulas (see Appendix A). While this approximation becomes accurate at the end of the reverse diffusion process (i.e. for low noise levels), it is less accurate at earlier stages where $p(x^{k+1} \mid x_t^{k+1}, x^k)$ is typically multimodal and far from Gaussian. An interesting direction for future work would be to rigorously evaluate different posterior sampling methods (Chung et al., 2023; Rozet et al., 2024; Zheng et al., 2025) on weather data using coverage and accuracy metrics (Lemos et al., 2023; Sharief et al., 2026).

As explained in Section 3, particle weights are also inexact, as they are obtained by approximating $p(x^{k+1} \mid x^k)$ with a Dirac centered at $\mathbb{E}\left[x^{k+1} \mid x^k\right]$. In preliminary experiments on Lorenz63, we found that approximate and exact weights (computed using Monte Carlo) lead to similar results when the number of particles is sufficiently large. However, in high-dimensional settings, the number of particles is typically small compared to the dimension of the system, making improved weight approximations an important avenue for future research.

Finally, although FA-APF uses the optimal proposal to minimize the variance of particle weights, inflation of the observation covariance matrix remains necessary in high-dimensional systems to prevent weight collapse. Although this biases the approximation of the filtering distribution by modifying the likelihood, it leads to strong empirical results while preserving particle diversity.

