# OpenReview forum: "Training-Free Bayesian Filtering with Generative Emulators"
_ICML.cc/2026/Conference — ICML 2026 spotlight_

### Official Review · Reviewer_Db5Q · 2026-03-04

**Soundness:** 3
**Presentation:** 3
**Significance:** 3
**Originality:** 3
**Overall Recommendation:** 4
**Confidence:** 4

**Summary:**

This work proposes a framework that utilizes a pretrained generative diffusion emulator, $p_\theta(x^{k+1}|x^k)$, for particle filtering—specifically, sampling from $p(x^k|y^1,...,y^k)$—without requiring additional training. The framework is structured as follows:

1. It builds upon the Fully-Adapted Auxiliary Particle Filter algorithm, which assumes access to a perfect proposal distribution $p(x^{k+1}|x^k, y^{k+1})$.
2. To sample from this perfect proposal, the authors assume the diffusion emulator is precise (i.e., $p_\theta(x^{k+1}|x^k) \approx p(x^{k+1}|x^k)$) and apply Bayes' rule, a standard technique in conditional diffusion generation. This yields the score of $p(y^{k+1}|x_t^{k+1}, x^k)$, which is akin to a classifier score in conditional diffusion, where $y^{k+1}$ acts as the condition.
3. In this particular particle filtering setting, $p(y^{k+1}|x^{k+1})$ is a known Gaussian. The authors decompose $p(y^{k+1}|x_t^{k+1}, x^k)$ by marginalizing the product of the kernels $p(y^{k+1}|x^{k+1})$ and $p(x^{k+1}|x_t^{k+1}, x^k)$. The latter is approximated as a Gaussian characterized by first- and second-order Tweedie formulas.
4. Finally, the algorithm calculates importance weights for resampling, which depend on an intractable term $p(y^{k+1}|x^k)$. The authors approximate this with a Gaussian, $p(y^{k+1}|\hat{x}^{k+1})$, where $\hat{x}^{k+1}$ is the denoising mean given by the diffusion model conditioned on $x^k$.

Overall, the paper proposes an intuitive and mathematically sound framework backed by strong experimental results that showcase its effectiveness. The primary concerns lie in (1) the multiple approximations used for tractability, as some are known to introduce inaccuracies (detailed in the weaknesses section), and (2) the computational overhead introduced by Eq (21) which requires the Jacobian of the denoiser network. Given the soundness of the empirical results and the creativity of the approach, I recommend a weak accept. Including a theoretical analysis to justify the errors introduced by these approximations would elevate this to a solid accept.

**Compliance With Llm Reviewing Policy:**

Affirmed.

**Final Justification:**

I thank the author for their effort on clarifying the problems. Most of my concerns are solved. The only left one is regarding the inference time in high dimensional systems, where the author claimed "this gap disappears in high-dimensional systems, where numerical solvers become extremely costly." but they didn't show any numerics justifying this statement. It is important to see how efficient or expensive it is in large systems.

Overall, I prefer to maintain my score as a weak accept.

**Key Questions For Authors:**

1. Could you elaborate on the errors introduced by the various approximations in the pipeline? A high-level discussion identifying which approximations are most critical to performance would be helpful. For those that could introduce large mismatches (e.g., the Gaussian approximation at large noise levels), could you provide a more detailed theoretical error analysis? Addressing this would significantly strengthen my evaluation of the paper's soundness.
2. Could you provide a detailed breakdown of the inference time and memory usage, particularly in comparison to the baselines? Specifically, I am interested in the overhead of differentiating the "classifier" in Eq. 21 (which is fairly noted in the limitations) and computing the Jacobian for the second-order Tweedie formula. Clarifying the practical scalability of these steps would help alleviate concerns regarding computational overhead.

**Limitations:**

Yes

**Strengths And Weaknesses:**

# **Soundness**
The proposed method is technically sound. Although the pipeline relies on multiple mathematical approximations, it achieves strong empirical performance across various benchmarks. A deeper theoretical justification of the approximation errors (discussed below) would further improve the soundness of the claims.

# **Presentation**
The paper is largely well-written, with a clear background introduction and motivation. However, clarity drops slightly around L174-L180 (left column) when introducing the score of the posterior.
* *Suggestion:* To improve clarity, I recommend using $\nabla\log$ instead of $s$ for the scores. Additionally, for notational simplicity, it would be helpful to let $(x, y)$ denote the current pair and $\tilde{x}$ denote the future prediction.

# **Significance**
The paper addresses a relevant problem and demonstrates practical utility. However, there are notable concerns regarding the robustness of the approximations and the computational overhead:
1. On the approximation of $p(x^{k+1}|x_t^{k+1}, x^k)$: The authors approximate this with a Gaussian characterized by the denoising mean and variance. However, at large noise levels, the true denoising distribution is often multimodal and far from Gaussian, which can introduce significant errors. While the experimental results are positive, it is necessary to discuss and characterize this error. The same applies to other approximations in the paper that leverage the denoising mean.
2. On the computational overhead: Employing the second-order Tweedie formula requires calculating the Jacobian of the denoiser, which is generally infeasible in high-dimensional settings. The paper lacks details on how this is overcome in practice, nor does it explicitly discuss the actual computational overhead incurred.

# **Originality**
The framework exhibits a moderate degree of novelty. While several of the underlying techniques are already established in diffusion-based posterior sampling and conditional generation tasks, creatively combining this domain knowledge to advance particle filtering is an original and valuable contribution.

---

> ### Author Rebuttal · Authors · 2026-03-30
>
> Thank you for your in-depth review and the pertinent questions you have asked. We hope that the following answers will address your questions.
>
> ## Weaknesses
>
> > The authors approximate this with a Gaussian characterized by the denoising mean and variance.
>
> The approximation used for $p(x^{k+1} \mid x^{k+1}\_{t}, x^{k})$ follows from the MMPS assumption that the prior is Gaussian. As the reviewer correctly points out, this assumption may be inaccurate at early diffusion steps with high noise levels, but becomes more reasonable at later steps of the reverse diffusion process.
>
> However, it is important to note that our method, and in particular line 13 of Algorithm 1, can be applied with any posterior sampling method. Therefore, any improved variant (using a Gaussian mixture prior for example) could be seamlessly incorporated into our algorithm as a replacement for MMPS, and we believe that evaluating and refining posterior sampling methods is beyond the scope of this work.
>
> > Employing the second-order Tweedie formula requires calculating the Jacobian of the denoiser, which is generally infeasible in high-dimensional settings.
>
> We agree that using the second-order Tweedie's formula in practice is not trivial, and we apologize if it is unclear.
>
> In high-dimensional settings, the challenge lies in storing (and using) the full covariance matrix rather than computing the denoiser’s Jacobian. For instance, with $x^{k}$ of dimension $10^{6}$ (as in GenCast), storing the covariance would require ∼4 TB of memory, and using it to solve Eq. (20) would then be expensive. Fortunately, because the formula involves the Jacobian of the denoiser, one can compute covariance–vector products efficiently, allowing us to compute the term $(\Sigma_{y} + \mathrm{H}\mathrm{V}\mathrm{H}^{\top})^{-1}\left[y^{k+1} - \mathcal{H}(m) \right]$ in Eq. (20).
>
> More precisely, the term $(\Sigma_{y} + \mathrm{H}\mathrm{V}\mathrm{H}^{\top})^{-1}\left[y^{k+1} - \mathcal{H}(m) \right]$ can be expressed as the solution of the linear problem $Ax = b$ with $A = (\Sigma_{y} + \mathrm{H}\mathrm{V}\mathrm{H}^{\top})$ and $b=y^{k+1} - \mathcal{H}(m)$. This problem can be solved using implicit solvers such as GMRES, which do not store $A$ explicitly but only require access to the matrix-vector operator $Ax$. In our case, using Torch’s jvp/vjp functions (see the code provided in the supplementary material), we can compute matrix-vector products between the Jacobian of the denoiser and any vector at (approximately) the cost of a pass through the denoiser, thus allowing us to define the $Ax$ operator and solving the system efficiently.
>
> ## Questions
>
> > Could you elaborate on the errors introduced by the various approximations in the pipeline?
>
> In low-dimensional tests on the Lorenz63 system (which may not fully generalize to high-dimensional cases), we mainly observed the following:
>
> - When the number of particles is sufficiently large, the metrics obtained using approximate weights are very close to those obtained with exact weights. With fewer particles, the metrics degrade slightly.
>
> - Even with exact weights and a large number of particles, we observed a small discrepancy between the true filtering distribution (obtained using BPF with a large number of particles) and the distribution produced by Algorithm 1. This is due to the error introduced by MMPS.
>
> - If the denoiser is well trained, expectation approximations are very accurate.
>
> These observations suggest that one of the most critical factors for improving the performance of our method is the posterior sampling algorithm used in line 13 of Algorithm 1. As noted previously, we currently use MMPS, but any other method could be integrated. Furthermore, in high-dimensional settings (where the number of particles is relatively small compared to the dimension of the system), our experiments indicate that weight approximation plays an important role. Therefore, a key direction for future work is to improve the computation of the weights.
>
> > Could you provide a detailed breakdown of the inference time and memory usage [...]?
>
> For the time complexity of the algorithm, due to characters limitations, we refer the reviewer to the detailed analysis provided to reviewers 3DMi and ZZym.
>
> Regarding the memory complexity, the use of automatic differentiation to compute jvp and vjp at each diffusion step requires storing the denoiser activations. For foundation models such as GenCast, this can be particularly demanding in terms of VRAM. However, we were able to run the 1° resolution model on H100 GPUs with 80GB of VRAM without relying on memory-saving techniques such as gradient checkpointing.
>
> As the complexity of the algorithm was also raised by other reviewers, we will include a detailed discussion in the revision.
>
>
> We hope that our detailed responses address your concerns and, if so, we would be very grateful if you could elevate your recommendation to a solid accept.

---

> > ### Author Rebuttal · Reviewer_Db5Q · 2026-04-01
> >
> > Thank you for clarifying the approximation errors, I generally acknowledge the practical performance. However, it is still unclear to me how biased it can be in practice, especially for the MMPS used in large systems.
> >
> > Regarding the complexity, could you talk about the actual inference time compared to other baselines?

---

> > > ### Author Response · Authors · 2026-04-03
> > >
> > > Thank you for your thoughtful comments and for acknowledging the practical performance of our work.
> > >
> > > We understand your concern regarding the use of MMPS in large systems. However, quantifying the biases of such methods in high-dimensional problems is difficult, since the true distributions (diffusion prior and posterior) are unknown. Nevertheless, we note that MMPS has been successfully applied in other high-dimensional data assimilation works [1-2].
> > >
> > > Regarding the complexity, here are the actual inference time (in seconds) per filtering step on the Lorenz'63 system.
> > >
> > > |        | N=16  | N=32  | N=64  | N=128 | N=256 | N=1024 |
> > > |--------|-------|-------|-------|-------|-------|--------|
> > > |  BPF   | 0.011 | 0.011 | 0.013 | 0.014 | 0.02  | 0.04   |
> > > |  EnKF  | 0.011 | 0.011 | 0.012 | 0.015 | 0.02  | 0.04   |
> > > |  EnSF  | 0.07  | 0.1   | 0.2   | 0.6   | 2.88  | 40     |
> > > |  EnFF  | 0.03  | 0.04  | 0.08  | 0.21  | 1.03  | 14     |Z
> > > | FA-APF | 3.36  | 3.5   | 3.5   | 3.5   | 3.5   | 3.5    |
> > >
> > > Classical methods such as BPF and EnKF are faster, as they can rely on numerical solvers to propagate particles (i.e, solving Eq. (29) for Lorenz63), whereas FA-APF requires solving the reverse diffusion equation at each filtering step. However, this gap disappears in high-dimensional systems, where numerical solvers become extremely costly (see the discussion on inference time in [3]). In such cases, using the diffusion model to propagate particles is much more efficient, and all methods have a per-step complexity of approximately $\mathcal{O}(N \times T \times C_{\mathrm{step}})$, with $N$ the number of particles, $T$ the number of diffusion steps to solve Eq. (11) and $C_{\mathrm{step}}$ the cost of a single diffusion step.
> > >
> > > In summary, our method is not substantially more expensive in high-dimensional settings. It can, however, be more costly when numerical solvers are fast, since FA-APF must solve the reverse diffusion equation, whereas other training-free methods can rely on the numerical solver. The table above is therefore not a strict apples-to-apples comparison, but highlights that our method is designed for generative emulators.
> > >
> > >
> > > [1] Andrae et al., DAISI: Data Assimilation with Inverse Sampling using Stochastic Interpolants, 2025.
> > >
> > > [2] Andry et al., Appa: Bending Weather Dynamics with Latent
> > > Diffusion Models for Global Data Assimilation, Machine Learning
> > > and the Physical Sciences Workshop (NeurIPS), 2025.
> > >
> > > [3] ECMWF Newsletter 181, Data-driven ensemble forecasting with the AIFS, Autumn 2024
> > >
> > > PS: We have slightly updated this comment to include reference [3], which supports the claim that classical numerical solvers are more costly than diffusion-based methods in high-dimensional settings.

---

### Official Review · Reviewer_ZZym · 2026-03-11

**Soundness:** 3
**Presentation:** 4
**Significance:** 3
**Originality:** 2
**Overall Recommendation:** 5
**Confidence:** 4

**Summary:**

The paper implements a diffusion model to generate proposal samples (particles) within a particle filtering framework. This allows the PF to sample from an approximation of the optimal proposal density.

**Compliance With Llm Reviewing Policy:**

Affirmed.

**Final Justification:**

Thanks for the response. I maintain my acceptance recommendation.

**Key Questions For Authors:**

In addition to my criticisms above, please refer to
- the computational complexity of the method.
- the required number of particles (and how that has an effect on the computational complexity)

**Limitations:**

Yes, they are identified in the paper (Section 7)

**Strengths And Weaknesses:**

The paper is very well presented, structured with an appropriate account of previous works and a clear experimental validation. Though one might argue that the idea is not radically novel (that has been an issue lately in ML conferences), as the paper puts together two well-studied objects, namely particle filters (or sequential Monte Carlo) and a diffusion model, the work is well elaborated around a clear objective: to sample from an approximation of the optimal proposal density. The experiments are well presented: two synthetic examples and one arguably real-world climate dataset, which is achieved by a foundation model, so it remains unclear whether that is a truly real-world data experiment.

My main criticism, which is also somewhat addressed in the paper as a limitation, is the fact that the method requires real (filtered) data so that the diffusion model (DM) can be trained, or a previously trained DM. This might be unrealistic in practice, and thus, this framework might be constrained to synthetic data or the cases where clean data, in the particular format that the model needs, are available. This is also in contrast with the title of the article, which reads "training-free...", but that is not the case if the DM needs to be trained on a previously unaddressed setting.

---

> ### Author Rebuttal · Authors · 2026-03-30
>
> Thank you for your review. We hope that the following answers will address your questions.
>
> ## Weaknesses
>
> > Though one might argue that the idea is not radically novel [...], as the paper puts together two well-studied objects, namely particle filters (or sequential Monte Carlo) and a diffusion model, [...].
>
> We agree that the proposed method is not fundamentally new, as it shows that generative emulators can be used without additional training to implement an existing optimal particle filter variant (FA-APF) which has received relatively little attention in the data assimilation community. Nevertheless, beyond establishing an elegant connection between generative models and data assimilation, our work demonstrates that the FA-APF remains competitive with other filtering algorithms and can be applied in high-dimensional settings. We hope that this study will pave the way for practical applications and further investigations of the FA-APF in operational scenarios.
>
> > [...] the method requires real (filtered) data so that the diffusion model (DM) can be trained, or a previously trained DM. This might be unrealistic in practice, [...].
>
> It is true that training a generative forecasting model requires realistic data from a dynamical system of interest. Nevertheless, this is not unrealistic in practice, as several domain-specific datasets already exist, where trajectories have been reconstructed a posteriori using real observations, physical laws, and statistical post-processing. Notable examples include ERA5 [1], which provides hourly atmospheric data since 1940, and Glorys [2], which provides daily oceanographic data.
>
>
>
> ## Questions
>
> > the computational complexity of the method.
>
> Using the notation of the paper, let $K$ denote the total number of filtering steps, $N$ the number of particles, $T$ the number of diffusion steps required to solve Eq. (11), and $C_{\mathrm{step}}$ the cost of a single diffusion step. As the cost is dominated by the posterior sampling step (line 13 of Algorithm 1), the overall complexity of the algorithme is then $\mathcal{O}\left( K \times N \times T \times C_{\mathrm{step}} \right)$.
>
> In our implementation, posterior sampling is performed using MMPS [3]. This requires solving a linear system iteratively using GMRES [4], which involves vector–Jacobian product evaluations and leads to a cost of approximately $C_{\mathrm{step}} = \mathcal{O}\left(M \times C_{\mathrm{denoiser}} \right)$, where $M$ is the number of GMRES iterations and $C_{\mathrm{denoiser}}$ the cost of a pass through the denoiser. The resulting total complexity is therefore $\mathcal{O}(K \times N \times T \times M \times C_{\mathrm{denoiser}})$.
>
> Importantly, our method is compatible with any posterior sampling procedure, including less expensive alternatives, although we adopt MMPS here due to its superior empirical performance. Also, the posterior sampling step is parallelizable across particles, which can reduce the effective filtering cost by a factor $N$.
>
> As the complexity of the algorithm was also raised by other reviewers, we will include a detailed discussion in the revision.
>
> > the required number of particles (and how that has an effect on the computational complexity)
>
> The number of particles required is difficult to quantify, as it depends on multiple factors such as the quality and quantity of observations, the temporal resolution, the system dimensionality, and its degree of chaoticity. In practice, one typically uses as many particles as permitted by the computational budget. In our experiments, we achieved filtering over 15-day trajectories with GenCast using 256 particles, with posterior sampling (line 13 of Algorithm 1) parallelized across 8 H100 GPUs.
>
>
> ### References
>
> [1] Hersbach et al., The ERA5 global reanalysis, Quarterly Journal of the Royal Meteorological Society, 2020.
>
> [2] Lellouche et al., The Copernicus Global 1/12° Oceanic and Sea Ice GLORYS12 Reanalysis, Frontiers in Earth Science, 2021.
>
> [3] Rozet et al., Learning Diffusion Priors from Observations
> by Expectation Maximization, The Thirty-eighth Annual Conference on Neural Information Processing Systems, 2024.
>
> [4] Saad et al., GMRES: A Generalized Minimal Residual Algorithm for Solving Nonsymmetric Linear Systems, SIAM Journal on Scientific and Statistical Computing, 1986.

---

> > ### Author Rebuttal · Reviewer_ZZym · 2026-04-04
> >
> > Thanks for the response. I maintain my acceptance recommendation.

---

> > > ### Author Response · Authors · 2026-04-07
> > >
> > > Thank you for your positive assessment of our work. We will carefully include a discussion on the complexity of our method in the revision.

---

### Official Review · Reviewer_3DMi · 2026-03-12

**Soundness:** 4
**Presentation:** 4
**Significance:** 4
**Originality:** 3
**Overall Recommendation:** 5
**Confidence:** 4

**Summary:**

This paper proposes a method for Bayesian filtering in systems with complex dynamics, replacing traditional PDE simulators with a DNN generative model. It uses a fully adapted auxiliary particle filter (FA-APF) which requires sampling from $p(x^{k+1} | x^k, y^{k+1})$. The proposed method does this using a denoising diffusion model that is trained conditional on $x^k$, together with tricks to approximate the score $\nabla_{x_t^{k+1}} p(x^{k+1}\_t | x^k, y^{k+1})$ in terms of $\\nabla\_{x_t^{k+1}} p(x^{k+1} | x^k)$ (as estimated by the diffusion model) and the observation model $p(y^{k+1}|x^{k+1}) = \mathcal{N}(y^{k+1}|\mathcal{H}(x^{k+1}),\Sigma_y)$.

**Compliance With Llm Reviewing Policy:**

Affirmed.

**Final Justification:**

Method is novel, theoretically sound, and a clever synergy between particle filters and generative diffusion. Empirical results are strong. My concerns were mostly superficial and nearly all addressed.

**Key Questions For Authors:**

In going from (19) to (20) it looks like you’re treating $\frac{\partial H}{\partial m}$ as negligible, i.e. approximating $\mathcal{H}$ as linear. Is that correct?

Isn't sec 3.3 redundant with the earlier decision to use FA-APF? That is, FA-APF basically amounts to replacing (7) with lines 5-10 of algo 1, so it was a little confusing to read a discussion starting with the question of how to compute (7) and concluding with lines 5-10. (Eqs 27-28 are useful to point out though.)

(1) abuses notation but also that notation ($\mathcal{M},\eta$) is not used again

There are many misspellings and other typos

**Limitations:**

yes

**Strengths And Weaknesses:**

Strengths
* Clever integration of particle filters and diffusion models: the score function calculated by the model allows sampling from the posterior $p(x^{k+1}|x^k,y^{k+1})$ even though the model is only trained on the marginal $p(x^{k+1}|x^k)$.
* Strong performance over baselines, often with a smaller number of particles

Weaknesses
* No formal analysis (theoretical or experimental) of compute cost, which the paper notes is high

---

> ### Author Rebuttal · Authors · 2026-03-30
>
> Thank you for your review and reading the manuscript in detail. We hope that the following answers will address your questions.
>
> ## Weaknesses
>
> > No formal analysis (theoretical or experimental) of compute cost, which the paper notes is high.
>
> Indeed, the original manuscript states that the computational cost is high, but does not provide further details. Using the notation of the paper, let $K$ denote the total number of filtering steps, $N$ the number of particles, $T$ the number of diffusion steps required to solve Eq. (11), and $C_{\mathrm{step}}$ the cost of a single diffusion step. As the cost is dominated by the posterior sampling step (line 13 of Algorithm 1), the overall complexity of the algorithm is then $\mathcal{O}\left( K \times N \times T \times C_{\mathrm{step}} \right)$.
>
> In our implementation, posterior sampling is performed using MMPS [1]. This requires solving a linear system iteratively using GMRES [2], which involves vector–Jacobian product evaluations and leads to a cost of approximately $C_{\mathrm{step}} = \mathcal{O}\left(M \times C_{\mathrm{denoiser}} \right)$, where $M$ is the number of GMRES iterations and $C_{\mathrm{denoiser}}$ the cost of a pass through the denoiser. The resulting total complexity is therefore $\mathcal{O}(K \times N \times T \times M \times C_{\mathrm{denoiser}})$.
>
> Importantly, our method is compatible with any posterior sampling procedure, including less expensive alternatives, although we adopt MMPS here due to its superior empirical performance. Furthermore, the posterior sampling step is parallelizable across particles, which can reduce the effective filtering cost by a factor $N$.
>
> As the complexity of the algorithm was also raised by other reviewers, we will include a detailed discussion in the revision.
>
>
> ## Questions
>
> > In going from (19) to (20) it looks like you’re treating $\frac{\partial H}{\partial m}$ as negligible, i.e. approximating $\mathcal{H}$ as linear. Is that correct?
>
> Originally, MMPS [1] was developed for linear observation operators $H$, in which case the score of the posterior (Eq. (20) in [1]) reduces to the gain of the standard Kalman Filter (see Eq. (4.21), p. 57 in [3]). In our setting, the observation operator may be nonlinear, so we instead rely on the Extended Kalman Filter, which linearizes the observation operator around the mean as
>
> $$ \mathcal{H}(x^{k+1}) \approx \mathcal{H}(m) + \mathrm{H}(x^{k+1} - m) $$
>
> with $m = \mathbb{E}[x^{k+1} \mid x^{k+1}\_{t}, x^{k}]$ and $\mathrm{H} = \left. \frac{\partial \mathcal{H}}{\partial x} \right|\_{x = m}$, leading to Eq. (19) (see Eq. (5.27), p. 70 in [3]). Taking the gradient of the logarithm of (19), under the assumption that the derivative of $\mathbb{V}[x^{k+1} \mid x^{k+1}\_{t}, x^{k}]$ with respect to $x^{k+1}\_{t}$ is negligible, then gives Eq. (20).
>
> > Isn't sec 3.3 redundant with the earlier decision to use FA-APF? [...]
>
> Section 3.3 aims to explain how the weights are computed in the case of FA-APF, and more specifically how lines 5–10 of Algorithm 1 are derived from Eq. (7), which is non-trivial and requires additional details. First, we highlight that Eq. (7) simplifies to Eq. (22) for FA-APF. Then, we describe how Eq. (22) can be efficiently approximated using the pre-trained denoiser.
>
> > (1) abuses notation but also that notation $(\mathcal{M}, \eta)$ is not used again.
>
> We agree that the notations in Eq. (1) are not used again in the paper, but we have included them to align with the notations commonly used in the data assimilation community.
>
> > There are many misspellings and other typos.
>
> We thank the reviewers for pointing out the typos in the manuscript. These have now been corrected, and we hope they did not hinder the readability of our work.
>
>
> ### References
>
> [1] Rozet et al., Learning Diffusion Priors from Observations
> by Expectation Maximization, The Thirty-eighth Annual Conference on Neural Information Processing Systems, 2024.
>
> [2] Saad et al., GMRES: A Generalized Minimal Residual Algorithm for Solving Nonsymmetric Linear Systems, SIAM Journal on Scientific and Statistical Computing, 1986.
>
> [3] Simo Särkkä, Bayesian Filtering and Smoothing, Cambridge University
> Press, 2013.

---

> > ### Author Rebuttal · Reviewer_3DMi · 2026-04-02
> >
> > The complexity discussion is a good addition.
> >
> > The explanation about linearizing $\mathcal H$ was what I expected but should be added to the paper.
> >
> > I still think sec 3.3 is introduced strangely but this is a matter of style that I leave to the authors. (The ingredient needed for FA-APF is the predictive mean in alg 1 line 5 and eq 27-28. Given that value, FA-APF computes its own weights. Eq 22-26 explain why those weights agree with eq 7, i.e., why FA-APF works, which I agree is helpful.)
> >
> > Using $\mathcal{M},\eta$ is fine but eq 1 is still nonsensical. It equates a distribution with a random variable. Correct would be to swap the order, $x^{k+1} = \mathcal{M}(x^k,\lambda)+\eta^{k+1}\sim p(x^{k+1}\vert x^k)$, or to write separate equations: $x^{k+1}\sim p(x^{k+1}\vert x^k)$ and $x^{k+1} = \mathcal{M}(x^k,\lambda)+\eta^{k+1}$.
> >
> > I noticed a typo at 127r: ${\rm N_{thr}^{min},N_{thr}^{min}}$

---

> > > ### Author Response · Authors · 2026-04-03
> > >
> > > We thank the reviewer for their valuable comments, which help improve the clarity and overall quality of the manuscript. In the revision, we will add discussions on the computational complexity and the linearization of $\mathcal{H}$. We will also replace Eq. (1) with $x^{k+1}=\mathcal{M}(x^{k}, \lambda) + \eta^{k+1} \sim p(x^{k+1} \mid x^{k})$, in accordance with the reviewer's suggestion.

---

### Official Review · Reviewer_fzoj · 2026-03-12

**Soundness:** 4
**Presentation:** 4
**Significance:** 3
**Originality:** 3
**Overall Recommendation:** 5
**Confidence:** 4

**Summary:**

This work researches Bayesian filtering in nonlinear dynamical systems using diffusion-based generative emulators. The main idea is to repurpose such emulators to approximate a fully adapted auxiliary particle filter (FA-APF) without additional training. The method uses MMPS to approximately sample from the optimal proposal $p(x^{k+1}\mid x^k,y^{k+1})$ by combining the prior score from the denoiser with an approximate likelihood score, and then computes particle weights using a tractable approximation of $p(y^{k+1}\mid x^k)$. Experiments are reported on Lorenz'63, 2D Navier--Stokes, and medium-range weather forecasting with GenCast.

**Compliance With Llm Reviewing Policy:**

Affirmed.

**Final Justification:**

My final recommendation remains positive. The paper is original, practically significant, and empirically strong in showing how pretrained generative emulators can be repurposed to implement an approximate FA-APF-style filter that scales from low-dimensional benchmarks to large weather settings, while the rebuttal helpfully clarified the approximate nature of the proposal sampling, weight computation, and inflation steps.

Although my concerns about cumulative bias, approximation quality, and sensitivity are only partially resolved, the authors acknowledged these limitations clearly and indicated they will sharpen the discussion in the revision; overall, this reinforced rather than weakened my view that the paper makes a technically solid and interesting contribution.

**Key Questions For Authors:**

1. can you provide an ablation for the Dirac-mass approximation in the weights, especially in settings where transition uncertainty is large?

2. how sensitive are the results to the inflation parameters, and when does inflation dominate the posterior approximation?

3. how the different approximation errors compound, e.g. whether poor weights trigger stronger inflation and further bias the filter?

4. for GenCast, do the reported gains remain when averaged over a larger and more diverse set of reference trajectories?

**Limitations:**

The method depends strongly on the quality of the pretrained generative emulator. More importantly, the practical algorithm is an approximation to the ideal FA-APF, because both proposal sampling and weight computation are approximate and additional inflation is used. The paper demonstrates promising empirical behaviour, but the cumulative bias from these approximations is not characterised.

**Strengths And Weaknesses:**

strengths:
1. reuse of pretrained generative models. The approach does not require retraining the diffusion emulator. The prior score comes directly from the denoiser, and the same model is reused for approximate proposal sampling and weight computation.

2. particle filtering ground. Unlike purely generative rollout methods, the proposed method keeps the auxiliary particle filter structure and explicit resampling/weighting steps. This gives it a clearer Bayesian interpretation than methods that ignore particle weights entirely.

3. Empirically sound. It evaluates the method from low-dimensional chaos to Navier--Stokes and a large-scale GenCast setting, which makes the work practically interesting.

weaknesses:
1. approximate proposal sampling. The MMPS step relies on a Gaussian approximation for the intermediate conditional density in Eqs.(18)-(20). This is reasonable as a practical approximation, but its error is not quantified and may be problematic in highly nonlinear regimes.

2. approximation in weight computation. In Sec.3.3 the transition density is replaced by a Dirac mass at the conditional mean, yielding Eqs.(23)-(28). This makes the weights tractable, but collapses transition uncertainty in the weight computation and may be unreliable when the true transition could be broad or multimodal.

3. inflation introduces additional bias. Variance inflation introduces bias, even if it is helpful in practice to avoid particle collapse. This weakens the claim of strict adherence to the exact FA-APF posterior.

4. accumulated error. The method combines approximate posterior-score evaluation, approximate weight computation, and inflation. The paper does not analyze how these errors accumulate.

---

> ### Author Rebuttal · Authors · 2026-03-30
>
> Thank you for your in-depth review. We hope that the following answers will address your concerns.
>
> ## Weaknesses
>
> > Approximate proposal sampling.
>
> Indeed, the posterior sampling method used in this work (MMPS [1]) is not exact, primarily because it assumes that the prior $p(x^{k+1} \mid x^{k+1}\_{t}, x^{k})$ is Gaussian, which is inaccurate at early stages of the reverse diffusion process, but becomes more reasonable later in the process.
>
> However, it is important to note that our method, and in particular line 13 of Algorithm 1, can be executed with any posterior sampling method, allowing it to benefit from future advances in posterior sampling techniques.
>
> > Approximation in weight computation.
>
> Computing the weights for FA-APF is non-trivial, as it requires approximating $p(y^{k+1} \mid x^{k})$ for each particle. In practice, we do so by approximating $p(x^{k+1} \mid x^{k})$ with a Dirac at $\mathbb{E}\left[x^{k+1} \mid x^{k} \right]$. While effective in practice, we fully agree with the reviewer that this is a rough approximation, and improving weight computation is an important avenue for future work.
>
> > Inflation introduces additional bias.
>
> Indeed, using an inflation coefficient in Algorithm 1 biases the estimation of the filtering distribution, but also helps control the spread of the distribution, keeping the spread-to-skill ratio close to 1 and not too low.
>
>
> ## Questions
>
> > Can you provide an ablation for the Dirac-mass approximation in the weight [...] ?
>
> In practice, we observed across the different problems that, when the denoiser $d_{\theta}$ is well trained, the approximation of $\mathbb{E}\left [ x^{k+1} \mid x^{k} \right]$ by $d\_{\theta}\left(x^{k+1}\_{t=1} = \sigma_{1}\varepsilon, x^{k}, t = 1 \right)$ is accurate.
>
> Also, in low-dimensional tests on the Lorenz63 system (which may not fully generalize to high-dimensional cases), we observed that when the number of particles is sufficiently large, the metrics obtained using approximate weights are very close to those obtained with exact weights. However, with fewer particles, the metrics degrade slightly.
>
> As this point was also raised by another reviewer, we will include a detailed discussion in the revision.
>
> > How sensitive are the results to the inflation parameters [...] ?
>
> We thank the reviewer for bringing up a point that we had not explored in depth. If time allows, we will include in the revision additional experiments on Lorenz63 and/or Navier–Stokes to investigate how the algorithm’s performance depends on the interval $[N\_{\text{thr}}^{\text{min}}, N\_{\text{thr}}^{\text{max}}]$ (linked to the inflation coefficient $\alpha$).
>
> > How the different approximation errors compound [...] ?
>
> From our low-dimensional experiments on the Lorenz63 system, we observed that even with a very large number of particles and exact weights, there is still a small discrepancy compared to the true filtering distribution (obtained using BPF with many particles). As pointed out by the reviewer in the weaknesses, this is due to the inexact posterior sampling with MMPS.
>
> Also, as explained earlier, in high-dimensional settings, the number of particles is relatively small compared to the dimension of the system, making weight computation errors significant.
>
> Thus, improving both posterior sampling and weights computations is key direction for future work.
>
> > For GenCast, do the reported gains remain when averaged over a larger and more diverse set of reference trajectories?
>
> The results for GenCast section correspond to a single reference trajectory from ERA5 [2]. However, we have tested our method on several other simulated trajectories (generated by GenCast) and observed similar results. If time permits, we plan to conduct further experiments on additional ERA5 reference trajectories in order to average the results and compute standard deviations.
>
>
> ### References
>
> [1] Rozet et al., Learning Diffusion Priors from Observations
> by Expectation Maximization, The Thirty-eighth Annual Conference on Neural Information Processing Systems, 2024.
>
> [2] Hersbach et al., The ERA5 global reanalysis, Quarterly Journal of the Royal Meteorological Society, 2020.

---

> > ### Author Rebuttal · Reviewer_fzoj · 2026-04-02
> >
> > The rebuttal helpfully clarifies that, the method is an approximate but practically motivated FA-APF-style filter, and I appreciate the authors’ responses about the current limitations in proposal sampling, weight computation, and inflation. While my technical concerns about cumulative bias and sensitivity are only partially resolved, they do not change my overall positive view of the paper.

---

> > > ### Author Response · Authors · 2026-04-03
> > >
> > > We thank the reviewer for their thoughtful comments and positive feedback on our work. In Section 7 of the revision (Limitation & Future work), we will provide a clearer discussion of the current approximations in order to better identify areas for improvement.

---

### Decision · Program_Chairs · 2026-04-30

**Decision:**

Accept (spotlight)

**Comment:**

This submission addresses the problem of estimating unknown states in dynamical systems from observations using a combination of a fully adapted auxiliary particle filter and generative emulators. All reviewers praised both the mathematical and the empirical soundness. There were some concerns in the initial round of reviews about limitations that should be discussed in a revision, e.g., computational overhead or the Gaussianity of the proposals. However, these shortcomings have been addressed to the reviewers' satisfaction during the rebuttal period. Therefore, I recommend accepting this work.